# Preparation of Bovine Hides Gelatin by Ultra-High Pressure Technique and the Effect of Its Replacement Fat on the Quality and In Vitro Digestion of Beef Patties

**DOI:** 10.3390/foods12163092

**Published:** 2023-08-17

**Authors:** Mengying Liu, Yanlei Gao, Ruheng Shen, Xue Yang, Li Zhang, Guoyuan Ma, Zhaobin Guo, Cheng Chen, Xixiong Shi, Xiaotong Ma

**Affiliations:** College of Food Science and Engineering, Gansu Agriculture University, Lanzhou 730070, China; liumengying1860@163.com (M.L.); gyl19970925@163.com (Y.G.); ruhengshen@163.com (R.S.); yangxuelyh@163.com (X.Y.); maguoyuangsau@163.com (G.M.); guozhb007@163.com (Z.G.); chenchengmlj@163.com (C.C.); sxix77@163.com (X.S.); maxiaotong17@163.com (X.M.)

**Keywords:** cowhide gelatin, ultra-high-pressure technology, fat replacement, beef patties, quality, in vitro digestion

## Abstract

Beef skin gelatin can be used as a good substitute for animal fat in meat patties. In this paper, the effect of different parameters on low-fat beef patties with cowhide gelatin substituted for beef fat (0, 25%, 50%, 75%, 100%) prepared by ultra-high pressure assisted technology was investigated by texture, cooking loss, and sensory scores. The beef patties were also stored at 0–4 °C for 0, 7, 14, 21, and 28 d. The differences and changing rules of fatty acid and amino acid compositions and contents of beef patties with different fat contents were investigated by simulating gastrointestinal digestion in vitro. The optimal process formulation of low-fat beef patties with cowhide gelatin was determined by experimental optimization as follows: ultra-high pressure 360 MPa, ultra-high of pressure time of 21 min, NaCl addition of 1.5%, compound phosphate addition of 0.3%. The addition of cowhide gelatin significantly increased monounsaturated fatty acids, polyunsaturated fatty acids, amino acid content, and protein digestibility of beef patties (*p* < 0.05). Moreover, with the extension of storage time, the content of saturated fatty acids was significantly higher (*p* < 0.05), the content of monounsaturated and polyunsaturated fatty acids was significantly lower (*p* < 0.05), the content of amino acids was significantly lower (*p* < 0.05), and protein digestibility was significantly lower (*p* < 0.05) under all substitution ratios. Overall, beef patties with 75% and 100% substitution ratios had better digestibility characteristics. The results of this study provide a theoretical basis for gelatin’s potential as a fat substitute for beef patties and for improving the quality of low-fat meat products.

## 1. Introduction

Fat is a very important component in the processing of meat patties, containing a wide range of high-value and body-needed nutrients and significantly influencing the flavor, tenderness, and juiciness of meat products [1]. In addition to its pleasant sensory properties, it contains a large amount of essential amino acids, minerals, and vitamins [2]. However, high saturated fatty acid levels also lead to elevated low-density lipoprotein cholesterol (LDL-C) in the body [3] and an increased prevalence of chronic diseases such as obesity, coronary heart disease, and cardiovascular disease [4]. Therefore, research on fat substitutes and low-fat meat products has become a hot topic. The three main categories of ingredients currently used to replace animal fats are protein-based, fat-based, and carbohydrate-based. Vegetable oils have been shown to be directly used to replace animal fats in meat processing [5,6], but because vegetable oils contain high levels of unsaturated fatty acids and are liquid at room temperature, they can have adverse effects on meat product properties, texture, and oxidative stability [7]. Carbohydrate gels have similar lubricity and fluidity as animal fats, and meat products prepared by replacing animal fats have better cohesiveness and water retention, etc. [8]. However, these gels are more difficult to digest in the gastrointestinal tract and have disadvantages such as low freeze–thaw stability and low acid-heat stability [9,10]. Among them, protein-based gels can modify the physical and textural characteristics of meat products, reducing cooking losses and improving their nutritional properties [11,12]. 

Gelatin is collagen derived from animal skin, white connective tissue, and bone and is rich in essential amino acids [6]. It is an easily digestible protein that contains all essential amino acids except tryptophan [13]. Cowhide gelatin is a thermally reversible protein gel prepared by thermal denaturation and partial hydrolysis of cowhide collagen, which has a melting point close to human body temperature and melts rapidly upon entering the mouth [14]. To date, cowhide gelatin is widely used in the processing of a range of new products, such as meat patties, ice cream, and yogurt, to improve their textural and quality characteristics [15]. It has been found that the addition of cowhide gelatin significantly enhances the textural properties and dehydration shrinkage of yogurt [16], improves water retention and cooking loss of chicken [17], and helps confer the prospective bioactivities of skimmed yogurt [18]. In addition, livestock gelatin has a more stable gel structure compared to fish gelatin [19]. However, some scholars [18] found that the addition of cowhide gelatin delayed the fermentation of skimmed yogurt, reduced acid production, reduced interactions between casein molecules, and resulted in fewer protein gels, which reduced the texture (hardness, adhesion, and springiness) and stability of yogurt. They hypothesized that this may be closely related to covalent bonding, intermolecular forces, and structural properties within cowhide gelatin. Therefore, there is a need to further modify cowhide so that it can provide better product quality characteristics. Commonly used methods for preparing cowhide gelatin include acid, alkaline, and enzymatic methods, but these processes produce gelatin with a long lead time, low yield and quality, generate large amounts of waste, and acids and alkalis cause abnormal peptide bond breaks in collagen molecules, resulting in reduced cowhide gelatin quality [20]. Ultra-high pressure (UHP) technology is a food processing method with great potential in the food industry that is suitable for food preservation and protein-assisted extraction [21]. One study [22] found that the use of UHP had a significant effect on the structure of bullfrog skin collagen, improving the thermal stability of collagen, and that the technology is not dependent on the size, shape, or composition of the product and does not lead to significant nutrient loss, which is both safe and energy efficient.

It has also been found that gelatin also has potential applications in promoting gastrointestinal digestion, but most of the current research has focused on fish-derived materials. Carp gelatin hydrolysates and synthetic peptides have been shown by in vitro gastrointestinal digestion studies that common carp gelatin hydrolysates may be better alternatives to isolated peptides as ingredients in formulated antioxidant foods [23]. The results of gelatin and konjac gum binary hydrogel composition on gel formation, oral sensation, and gastrointestinal digestive properties showed that hydrogel composition could modulate gel swelling rate and digestive behavior, and enhanced gel strength could delay lipid digestion [24]. 

In recent years, it has been shown that cowhide gelatin can be used as a fat substitute to reduce the fat content of meat products. Based on the research hotspot of fat substitution, this study has deeply analyzed and investigated the gastrointestinal digestion and absorption characteristics of beef patties prepared by UHP modification technology, which is more accurate and reliable than the traditional method, and provided new ideas and inspiration for the research of fat-substituting food products in gastrointestinal digestion. Therefore, in this paper, different proportions of cowhide gelatin were used to replace animal fat to optimize the preparation process parameters of low-fat beef patties. By simulating the digestive system in vitro, the variation patterns of different fat content on fatty acid and amino acid composition, content, and digestibility of beef patties during refrigeration were investigated. It aims to evaluate the digestion and absorption characteristics of proteins and fats and to better understand the absorption mechanisms of amino acid and fatty acid molecules in the gastrointestinal tract.

## 2. Materials and Methods

### 2.1. Materials

Six healthy local crossbred cattle (Simmental × Gansu local Chinese Yellow Cattle), with an average age of 3 years, weighed between 400 and 450 kg. Humane slaughter was carried out at a commercial meat processing company (Tianzhu Huayue Star Cold Chain Logistics Co., Ltd., Wuwei, Gansu, China) in accordance with the National Standard of the People’s Republic of China, “Cattle Slaughtering Operation Procedures”. Immediately after slaughter, the cowhide was removed from the carcass. Over 95% of cowhide hair was successfully removed using a method widely used by the research group [25]. It was washed and stored at 4 ± 1 °C.

Beef hind leg meat, salt, starch, pepper, cloves, fennel, and star anise were purchased from the local Beijing Hualian supermarket. The phosphate complex was purchased from Jian Yin Food Technology Co. Ltd., (Shanghai, China). All other chemical reagents were analytical grade.

### 2.2. UHP-Assisted Preparation of Bovine Gelatin

The cowhides were cleaned, excess fat and muscle removed and cut into small pieces (1 cm × 1 cm). The cowhides were dried at 105 °C for 2 h and then defatted by Soxhlet extraction. Then five times the volume (*w*/*v*) of 1% NaCl solution was added to the hides, which were soaked for 12 h to remove non-collagenous material and rinsed three times with distilled water. The cowhide was mixed with distilled water (material-liquid ratio of 1:1.5) and vacuum packed in polyethylene bags (DZ-450, Wenzhou Dajiang Vacuum Packaging Machinery Co., Ltd., Wenzhou, Zhejiang Province, China). Using pure water as the medium, the packaged cowhide solution was pressurized under 200 min, 250 MPa, 300 MPa, 350 MPa, and 400 MPa pressure for 5 min, 10 min, 15 min, 20 min, and 25 min using UHP equipment (M3L/100, Changzhou Lanuo Optoelectronic Technology Co., Ltd., Changzhou, Jiangsu, China), while the temperature was maintained at 25 ± 2 °C using a cooling device. Both unpressurized and pressurized cowhide were heated at 105 °C for 6 h. The cowhide and other impurities were filtered repeatedly until the gel solution was transparent, cooled until the gel system was stable, and then freeze–dried and set aside.

### 2.3. Prepare Beef Patties Process

Fresh beef hind leg meat was selected, and the surface fascia and connective tissue, etc., were first removed, rinsed, cut into long strips, and twisted into minced meat using a meat grinder (AMG31B-160, Zhucheng Heyi Machinery Co., Ltd., Langfang, Shandong Province, China) with a sieve aperture of 8 mm, and the formulation of each sample is shown in Table 1. The pre-prepared cowhide collagen and beef fat were mixed in mass ratios of 0:1, 1:4, 1:1, 4:1, and 1:0, and NaCl, compound phosphate, and spices were added according to the proportions, stirred well, and marinated at 4 °C for 12 h. After that, starch and iced water were added and mixed well using a chopper (ZLQ, Anhui Valin Western Kitchen Equipment Co., Ltd., Langfang, Anhui Province, China). The well-mixed minced meat was shaped in a mold (8 cm in diameter and 1.5 cm in height) and fried at 110 °C for 5 min until both sides of the patties were golden brown, then removed (XML-81, Jiangmen Xinwenda Kitchen Equipment Co., Ltd., Jiangmen City, Guangdong Province, China). Finally, all meat patties were stored at 0–4 °C and analyzed on days 0, 7, 14, 21, and 28.

### 2.4. Experimental Design and Optimization

#### 2.4.1. Single-Factor Experiment

The single factor experiments were based on determining the ratio of kraft gelatin to beef fat of 1:4 and the addition of other excipients, selecting the UHP pressure of 200 MPa, 250 MPa, 300 MPa, 350 MPa, and 400 MPa, and the UHP time of 5 min, 10 min, 15 min, 20 min, and 25 min for the preparation of kraft gelatin. NaCl concentrations required for the preparation of beef patties were 1%, 1.3%, 1.5%, 1.7%, and 1.9%, and the amount of compound phosphate added was 0.1%, 0.2%, 0.3%, 0.4%, and 0.5%. The cooking loss, texture, and sensory score of beef patties were used as indicators to determine optimal process parameters for each factor.

#### 2.4.2. Box-Behnken Design

The response surface test was used to optimize the optimum process parameters of beef patties. On the basis of the one-way test, according to the Box–Behnken experimental design principle, UHP (300 MPa, 350 MPa, 400 MPa), UHP time (15 min, 20 min, 25 min), NaCl concentration (1.3%, 1.5%, 1.7%), and compound phosphate addition (0.2%, 0.3%, 0.4%) were selected as the investigated variables, and springiness and sensory score were the response values. Design-Expert 13.0 software was applied to design the response surface analysis test, and factor levels are shown in Table 2. There was a total of 29 test points, of which 24 were factors analyzed and 5 were nulls. The null test was performed six times, while the other tests were randomized and replicated three times. A quadratic equation was chosen to describe the relationship between factors and responses. Finally, a regression model was used to validate the surface response experiments.

### 2.5. Cooking Losses

Following the method of Moghtadaei, Soltanizadeh and Goli [7], the prepared patties were fried in rapeseed oil at 110 °C for 5 min. The weight before frying was noted as *m*_1_, and after frying, the patties were cooled to room temperature, and the weight was noted as *m*_2_. Three parallels were made for each treatment group, and the cooking losses were calculated as follows:(1)$$Cooking\;loss\;(\%)=\left(m_{1}-m_{2}\right)/m_{1}\times 100\%$$

### 2.6. Textural Analysis

According to da Silva, et al. [26], with minor modifications. The cooked patties were cooled to room temperature, cut into blocks (1 cm × 1 cm × 1 cm), and analyzed by texture profiling using the “TPA” mode of a texture profiler (TA.XT Express, Stable Micro Systems, UK). Measurement parameters: pre-test rate: 1.0 mm/s; test rate: 2.0 mm/s; post-test rate: 1.0 mm/s; interval between secondary pressurization 5 s, deformation: 50%; trigger value: 5 g; probe type: P36 (36 mm diameter). Each sample group was measured 12 times, and the mean value was taken.

### 2.7. Sensory Evaluation

Panel members (7 males and 7 females) with sensory evaluation experience were selected to perform sensory evaluation of beef patties. Meat patties were prepared in accordance with Section 2.3, and each sample was numbered using a three-digit random code and placed in a food tray for random assessment by panelists. Panel members had to rinse their mouths with purified water before evaluating each sample and were not allowed to communicate with each other during the evaluation. Items evaluated included color, taste, texture, and histomorphology. The sensory evaluation criteria are shown in Table 3.

### 2.8. Analysis of Fatty Acid Composition

The fatty acid composition of the samples was determined by gas chromatography-mass spectrometry using Xiong, et al. [27] with minor modifications. 0.6 g of the sample was weighed, and 0.4 mL of KOH solution (10 moL/L) and 3 mL of anhydrous methanol solution (undecanoic acid internal standard and methanol were mixed and prepared at a ratio of 1:200 (*w*/*v*)) were added. After shaking well, put in a water bath (HWS24, Shanghai Li-Chen Bang Xi Instrument Technology Co., Ltd., Shanghai, China) and heat at a constant temperature for 1.5 h (55 °C, shaking once at 20 min intervals), add 0.33 mL of concentrated H_2_SO_4_ solution, and heat in a water bath for 1.5 h (55 °C, shaking once at 20 min intervals). After adding 1.7 mL of hexane, centrifuge (TGL-16M, Changsha Xiangyi Centrifuge Instrument Co., Ltd., Changsha, Hunan Province, China) for 5 min (3000 rpm/min, 4 °C). The supernatant was filtered through an organic membrane (0.22 μm), and a gas chromatography-mass spectrometry (GC–MS) analyzer (7890B+5977, Shanghai Shanfu Electronic Technology Co., Ltd., Shanghai, China) was used to determine the compositions and contents of monounsaturated fatty acids (MUFA), polyunsaturated fatty acids (PUFA), and saturated fatty acids (SFA) in the beef patties before and after digestion with different fat contents.

### 2.9. Analysis of Amino Acid Composition

The amino acid composition and content were based on Tian, et al. [28] methods with slight modifications. The amino acid composition and content of beef patties with different fat content before and after digestion were determined by post-column derivatization analysis using a fully automated amino acid analyzer (L-8900, Shanghai Lundt Testing Instruments Co., Ltd., Shanghai, China). The determination was as follows: After mixing 25 mg of gelatin with 10 mL of superior pure hydrochloric acid in a hydrolysis tube with an alcohol blowtorch, corked and sealed with a heat-resistant membrane for closure seal, the hydrolysis solution was hydrolyzed in an oven (DZF-6020L, Shanghai Jinlan Instrument Manufacturing Company Limited, Shanghai, China) for 22–24 h (110 °C), cooled to room temperature, the hydrolysis solution was filtered, 1 mL of hydrolysis solution was accurately aspirated, and after the HCl acid was fully evaporated in an oven at 100 °C, 3 mL of deionized water was added, and the organic filter membrane (0.22 μm) was filtered and measured on the machine.

### 2.10. Digestibility

Simulated gastric fluid (SGF) and simulated intestinal fluid (SIF) for in vitro digestion were referred to by Gallego, et al. [29] with minor modifications. After all beef patties were cooked (decocted at 110 °C for 5 min), 2.00 g of ground samples were weighed and added to 8 mL of configured SGF (NaCl solution was first adjusted to a pH of 2 with HCl solution, followed by the addition of porcine pancreatic enzyme (21 U/mg) and pepsin (182 U/mg) and incubated at a constant temperature of 37 °C for 2 h. Then, the pH was adjusted to 7.0 ± 0.1 with NaOH solution, and the gastric fluid environment was formed. The pH of gastric fluid was continued to be adjusted to 7.5 ± 0.1, and 1.5 mL of prepared SIF (NaCl solution was firstly adjusted to pH 7.5 with NaOH solution, then α-pancreatic coagulation protease (0.44 U/mg), trypsin (34.5 U/mg), and 50 mg porcine pancreatic enzyme (200 U/mg)) were added and incubated at 37 °C in a water bath for 2 h. The final formation of an intestinal fluid environment. Anhydrous ethanol (1:3, *v*/*v*) was added to the digested hydrolysis products, and the precipitate was centrifuged at 10,000× *g* for 20 min (4 °C), and then the precipitate was vacuum freeze-dried (VFD-03, Dalian Fengzhou Technology Co., Ltd., Dalian, Liaoning Province, China) and set aside.

The in vitro digestibility (%) values were calculated according to the following equation:(2)$$digestibility(\%)=(P-P_{0})/P\times 100$$
where *P* and *P*_0_ represent the protein content in the sample before and after digestion, respectively.

### 2.11. Data Analysis

In this study, Design Expert 13.0 was used for the design and analysis of the response surface test, and all data were analyzed for significance by Duncan’s multiple range test with ANOVA, comparison of means, and Duncan’s multiple tests (*p* < 0.05) performed by SPSS 20.0 statistical software. Models were significantly different at *p* < 0.05. Each set of tests was repeated three times.

## 3. Results and Discussion

### 3.1. Optimization of Beef Patty Preparation Process

#### 3.1.1. Single-Factor Experimental Results

##### Effect of Bovine Collagen on Beef Patties under Different UHP

UHP-treated gelatin has greater aggregation and denaturation, which can improve its properties [30]. Meanwhile, the degree and type of conformational changes in collagen molecules can be controlled by adjusting the pressure [22]. Figure 1A,B show that the hardness, springiness, and sensory scores of beef patties increased significantly (*p* < 0.05) with the increase of UHP bovine collagen, and they all began to decrease when the pressure exceeded 350 MPa. Whereas the cooking loss was contrary to the above trend and reached a minimum of 350 MPa. The reason for this change may be due to the fact that the interaction between cowhide collagen and polysaccharides is strengthened with the increase of UHP pressure, the two polymer systems are closer together, and the interactions between non-covalent and hydrogen bonds are significantly enhanced, resulting in a more stable structure of the cowhide gel network [31]. However, protein denaturation is likely to be induced under excessive pressure, leading to significant disruption of the gel network and a corresponding reduction in the hardness and springiness of the patties [22]. In addition, suitable high pressure can promote the formation of a dense three-dimensional network structure in the collagen of beef hides, which contributes to more water being trapped in the network structure as bound water and thus can reduce the exudation of water from beef patties during the frying process [32]. However, when the pressure is too high, the fracture of the gel mesh structure will lead to the transfer of more free water from the internal to the external gel mesh structure, which increases the cooking loss of meat patties [33]. Combining the four evaluation indexes, UHP pressures of 300 MPa, 250 MPa, and 400 MPa were selected for response surface optimization.

##### Effect of Bovine Collagen on Beef Patties under Different UHP Time

It has been reported that UHP can disrupt the non-covalent binding equilibrium of proteins, which in turn induces protein denaturation, leading to changes in protein conformational and functional properties [32]. From Figure 2A,B, it can be seen that the hardness, springiness, and sensory scores of beef patties increased significantly (*p* < 0.05) with the extension of the UHP time of kraft collagen and began to decrease after more than 20 min. Cooking losses decreased with the extension of the UHP time of kraft collagen, reaching a low of 20 min, and showed an increasing trend thereafter. Previous studies have found that with the extension of UHP time, the secondary structure of cowhide collagen changed significantly, the triple helix structure was significantly loosened, and the content of macromolecular components was significantly increased, which enhanced the strength of cowhide collagen gels [25]. However, when the UHP time is too long, it induces protein deformation, breaks and destroys the gel network structure, and collagen gel strength decreases, leading to a decrease in the hardness and springiness of beef patties [31]. Combining the four evaluation indexes, UHP times of 15 min, 20 min, and 25 min were selected for response surface optimization.

##### Effect of Bovine Collagen on Beef Patties at Different NaCl Additions

In meat processing techniques, NaCl not only improves meat quality and water retention but also improves texture and enhances meat flavor [34]. As can be seen in Figure 3A,B, the springiness and sensory scores of beef patties increased by increasing NaCl content up to 1–1.5% NaCl addition, and both began to decrease after exceeding 1.5%. The hardness of beef patties tended to increase within 1–1.7% NaCl content and decreased significantly after exceeding 1.7% (*p* < 0.05). In addition, the cooking loss of meat patties continued to decrease with increasing NaCl content, and the reduction in cooking loss was not significant after exceeding 1.5% (*p* > 0.05). This may be due to the fact that NaCl promotes the solubilization of salt soluble proteins in beef patties, which improves the hardness and springiness of beef patties. Tobin, et al. [35] found that high NaCl concentrations may lead to a decrease in myofibrillar proteins in the protein gel network, an increase in water retention, and an excess of water content, making patties harder and less elastic. In addition, NaCl may also promote the dissociation of actinomyosin, leading to the full unfolding of the protein structure [34], where the movement of water molecules is bound to form a dense three-dimensional meshwork, which improves water retention and reduces the cooking loss of meatloaf [36]. Combining the four evaluation indexes, 1.3%, 1.5%, and 1.7% NaCl additions were selected for response surface optimization.

##### Effect of Kraft Collagen on Beef Patties with Different Compound Phosphate Additions

Studies have shown that complex phosphates have water retention, tenderization, antioxidant, and slight bacteriostatic effects on meat products [37]. As can be seen in Figure 4A,B, springiness and sensory scores of beef patties significantly increased and hardness decreased with the increase of complex phosphates, and they both changed slowly with the addition of more than 0.3% (*p* < 0.05). This is due to the fact that complex phosphates promote the dissociation of actinomyosin in meat patties to produce actin and myosin, and when the two are separated, water enters the interstitial space of muscle fibers [34], and tenderness increases, leading to a decrease in patties hardness and an increase in springiness [38]. In addition, beef patties had the highest sensory scores and the lowest cooking losses when complex phosphates were added at 0.3%. Jia, et al. [39] also found that complex phosphates significantly increased the pH value of fish fillets, which resulted in myofibrillar proteins deviating from the isoelectric point, repelling each other’s charges, and swelling proteins, which allowed fish tissues to hold more water and have higher water retention properties. However, when there is an excess of complex phosphates, it causes a change in protein conformation, leading to a decrease in fracture strength and deformation and a decrease in water retention, which increases the cooking loss of the patty [40]. Combining the four evaluation indexes, 0.2%, 0.3%, and 0.4% phosphate complex additions were selected for response surface optimization.

#### 3.1.2. Response Surface Optimization Test Results and Analysis of Variance

According to the results in Table 4, the data were analyzed by multiple regression fitting via Design-Expert 13.0 software, with UHP pressure A, UHP time B, NaCl addition C, and complex phosphate addition D as independent variables and the combined value of springiness (by weight 40%) and sensory score (by weight 60%) Y as dependent variables, and the quadratic multinomial equation was obtained as follows:Y = 0.99 + 3.165E − 003A + 1.636E − 003B + 3.999E − 003C − 1.728E − 004D + 3.576E − 003AB + 1.360E − 003AC + 3.990E − 003AD + 2.306E − 003BC + 1.720E − 003BD + 1.762E − 004CD − 9.062E − 003A^2^ − 0.012B^2^ − 0.012C^2^ − 0.010D^2^

The correlation coefficient R^2^ was 0.9704 and the correction coefficient R^2^adj for the experimental model was 0.9407, indicating that 97.04% of the variation in response values could be explained by the model and 94.07% of the experimental results were influenced by experimental factors, indicating that the data were reliable and that the regression model was a better fit and could be used for theoretical prediction of texture and sensory scores of beef patties.

### 3.2. Analysis of Variance

The actual test results were analyzed by ANOVA using a regression equation. The results are shown in Table 5, which shows that the model is highly significant (*p* < 0.0001) and the lost proposal is not significant (*p* > 0.05), indicating that the regression model is validly established. The effects of independent variables A and C on composite Y were highly significant (*p* < 0.01), B was significant (*p* < 0.05), and D was insignificant (*p* > 0.05). The interaction terms AB and AD in the model had a highly significant effect (*p* < 0.01) on the composite Y value, while AC, BC, BD, and CD had a non-significant effect (*p* > 0.05). The effects of the secondary terms A^2^, B^2^, C^2^, and D^2^ in the model on the composite Y value reached a very significant level (*p* < 0.01). In summary, the degree of influence of each factor on the Y composite value of springiness and sensory scores was A, C > B > D.

#### 3.2.1. Analysis of Interaction among All Factors

In order to more intuitively reflect the effect of the interaction between the four factors of UHP pressure (A), UHP time (B), NaCl addition (C), and compound phosphate addition (D) on the composite value Y, the response surface and contour plots of the relationship between each of the two factors and the composite value were plotted using Design Expert 13.0 software (Figure 5). The deeper the surface of the response surface and the closer the shape of the contour lines to an ellipse, the more significant the effect of the interaction between independent variables on the response value. Compared to the other plots, the contour lines of the AB and AD interaction terms (Figure 5A,C) are elliptical and densely distributed, and the surface is steeper, indicating that the interactions between UHP pressure and UHP time and compound phosphate, respectively, are obvious and have more significant effects on composite values (*p* < 0.01), which are consistent with what is described in Table 4.

#### 3.2.2. Validation Experiments

Through the regression model prediction (Figure 6), the optimal process conditions for optimizing low-fat beef patties with cowhide collagen assisted by UHP technology were determined to be: UHP pressure of 361.24 MPa, UHP time of 20.642 min, NaCl addition of 1.509%, and compound phosphate addition of 0.32%, at which time the sensory scores and elasticity of beef patties were 84.5% and 0.82%, respectively. Taking into account the possibility of practical operation, the optimal process conditions were adjusted to 360 MPa UHP pressure, 21 min UHP time, 1.5% NaCl addition, and 0.3% compound phosphate addition, the results showed that the sensory scores and elasticity of beef patties were 84.1% and 0.8%, respectively, as shown in Figure 6, and the validation was repeated three times, and the actual values were similar to the theoretical values of the model, which indicated that the model has some feasibility in optimizing the beef patty process.

### 3.3. Effect of Cow Skin Gelatin on Fatty Acids in Beef Patties

According to the recommendations of the Food and Agriculture Organization of the United Nations, unsaturated fatty acid/saturated fatty acid ratios less than 0.4 and n-6/n-3 ratios greater than 4 in meat and meat products are considered unhealthy for human diets because they are highly likely to promote cholesterolemia [41]. In the present study, 17 fatty acids, including 10 SFA, 4 MUFA, and 3 PUFA, were characterized in beef patties before and after digestion during refrigeration (Table 6). The results showed that SFA content in beef patties decreased significantly (*p* < 0.05) and MUFA and PUFA content increased significantly (*p* < 0.05) as fat substitution percentage increased. This result may be attributed to oxidative changes in PUFA due to the frying oil used [42]. In contrast to the other treatment groups, the control group (0%) meat patties had higher SFA content, especially C14:0 and C16:0, which are fatty acids characterized by high cholesterol and are strongly associated with chronic diseases such as obesity and coronary heart disease [43]. The SFA content of beef patties increased significantly (*p* < 0.05) and both PUFA and MUFA decreased significantly (*p* < 0.05) with increasing refrigeration time due to the high degree of unsaturation of the unsaturated fatty acids, which leads to the removal of protons and the generation of free radicals, which further oxidize to form SFA [44]. Although the MUFA content of beef patties all decreased during refrigeration, the decrease was less than that of PUFA due to the slower oxidation of MUFA than PUFA [45]. Furthermore, by comparing the fatty acid composition and content of beef patties before and after simulated gastrointestinal digestion (sGD), the total amount of SFA remained almost unchanged, the amount of MUFA increased significantly (*p* < 0.05), and the amount of PUFA decreased significantly (*p* < 0.05), which may be due to gastrointestinal conditions promoting lipid oxidation, which is greater with a greater number of double bonds [46]. Zhu, et al. [47], in a study on fatty acid release from emulsified lipids during in vitro digestion, noted that the structural composition of triglycerides and the length of their carbon chains determine the extent of fatty acid release. Overall, the fatty acid distribution in the digested patties was maintained intact due to the antioxidant activity of the bovine hide gelatin used in the beef patty formulation.

The main fatty acids in beef patties before and after digestion during refrigeration (Figure 7A) included C14:0 (myristic acid), C16:0 (palmitic acid), C18:0 (stearic acid), and C18:2n6c (linoleic acid). These fatty acid profiles were consistent with our previous studies in beef heart patties [25] and those reported by Lu, Kuhnle and Cheng [5] in pork patties. Based on cluster analysis, it can be seen (Figure 7B) that the fatty acid composition was divided into two major groups. The first group consisted of 10 SFA, 4 MUFA, and 2 PUFA; the second group consisted of C18:2n6c, PUFA, and MUFA; and the third group consisted of SFA. The percentage of fat substitution could be categorized into two main groups: the first group consisted of 100%, and the second group consisted of 0%, 25%, 50%, and 75%. In the present study, when more than 75% of the fat was replaced by cowhide gelatin, PUFA/SFA in meat patties before and after digestion were >0.4 and n-6/n-3 <4. Therefore, it is suggested that the use of cowhide gelatin as a fat substitute for beef patties may have a beneficial effect on maintaining the levels of essential fatty acids required for human health.

### 3.4. Effect of Cow Skin Gelatin on Amino Acids in Beef Patties

Beef patties are considered a good source of high-quality protein and also provide a balanced composition of essential amino acids [48]. As shown in Table 7, the amino acids determined in beef patties before and after in vitro sGD were Asp (Aspartic acid), Thr (Threonine), Vla (Valine), Ser (Serine), Glu (Glutamic acid), Gly (Glycine), Ala (Alanine), Cys (Cysteine), Tyr (Tyrosine), Phe (Phenylalanine), Lys (Lysine), His (histidine), and Pro (proline). The results showed that the amino acid content of digested beef patties increased significantly (*p* < 0.05) compared to predigestion as the percentage of substitution increased. This can be attributed to the fact that more enzymatic hydrolysis occurred in the samples digested with beef hide gelatin, and more free amino acids were produced as final products. Because the small intestinal transport mechanism only allows fewer amino acids, or short-chain peptides, than tetrapeptides, the end products of protein digestion have a beneficial effect on peptide absorption in the small intestine [49]. As storage time increased, cysteine content hardly changed, and the remaining amino acid content decreased significantly (*p* < 0.05). This may be due to high-temperature cooking, which increases protein oxidation in meat patties [50]. And with the significant increase (*p* < 0.05) in the content of most amino acids in digested beef patties, studies have shown that the increase in free amino acid content also contributes to the increase in antioxidant properties [51]. From the results of the increase in amino acids in this study, it can be inferred that the digestion of beef patties can increase the nutritional levels and physiological functions of the human body.

The main amino acids of beef patties before and after digestion (Figure 8A) included Asp, Thr, Glu, Cys, Lys, and His, which is consistent with the results in Table 7. Based on cluster analysis (Figure 8B), the relative composition of amino acids was divided into two major groups. The first category includes the amino acids Asp and Glu, and the above verified that Asp and Glu content tended to increase with the increase in the percentage of Kraft gelatin substitution. The second group consists of Tyr, His, and Lys, while the third group consists of Vla, Thr, Ser, Cys, Gly, Vla, Pro, and Phe. These results occur because pepsin plays an important role in the primary digestion of food proteins, producing long-chain peptides that are then digested by intestinal enzymes to form short-chain peptides and free amino acids. The availability of phenylalanine, tyrosine, tryptophan, and lysine as target cleavage sites for pepsin or trypsin determines the extent of protein digestion in the gastrointestinal tract [52].

### 3.5. Effect of Kraft Gelatin on the In Vitro Digestibility of Beef Patties Protein

Increased protein digestion facilitates hydrolysis and the production of short-chain peptides, which are essential for human metabolism [53]. As shown in Figure 9, protein digestibility increased significantly with the increase in cowhide gelatin content, which may be attributed to the increasing trend in protein digestibility due to the consequent increase in collagen content in beef patties with the increase in cowhide gelatin content (*p* < 0.05). In addition, Luo, et al. [54] found that higher protein concentrations and lower salt ion concentrations promoted protein–protein interactions and favored digestion into smaller peptides. However, protein digestibility was negatively correlated with refrigeration time. The digestibility of proteins tended to decrease with the increase in refrigeration time (*p* < 0.05). This may be due to changes in protein molecule structure in the early stages of oxidation resulting in exposure to most digestive enzyme binding sites, but with increased storage time, oxidation is enhanced, protein molecules continue to aggregate, and protein digestive enzyme binding sites are masked, resulting in decreased protein digestibility [55].

## 4. Conclusions

In summary, beef hide gelatin was prepared by UHP combined heating technology, and the modified gel with better properties was obtained by optimizing parameters and applied as a fat substitute in low-fat beef patties. In vitro gastrointestinal digestive characterization revealed good texture, sensory scores, and low cooking losses in beef patties with the addition of cowhide gelatin. The ratio of unsaturated to saturated fatty acids in beef patties with more than 75% substitution during refrigeration was consistent with the levels of essential fatty acids required for human health. And the free amino acid content of beef patties increased with the increase in gelatin substitution ratio, which significantly improved the digestibility of beef patties protein. The low-fat beef patties prepared in this study have good digestive properties and can enhance the nutritional level and physiological function of the human body.

## Figures and Tables

**Figure 1 foods-12-03092-f001:**
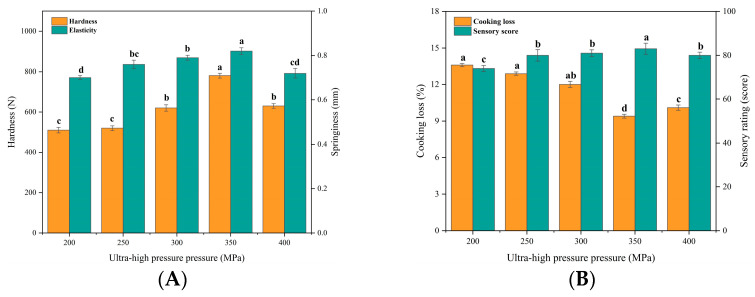
Effect of UHP pressure on texture, cooking loss, and sensory properties of beef patties. (**A**) Effect of UHP pressure (MPa) on hardness (N) and springiness (mm) of beef patties. (**B**) Effect of UHP (MPa) on cooking loss (%) and sensory characteristics (score) of beef patties. Different letters (a–d) indicate significant differences (*p* < 0.05).

**Figure 2 foods-12-03092-f002:**
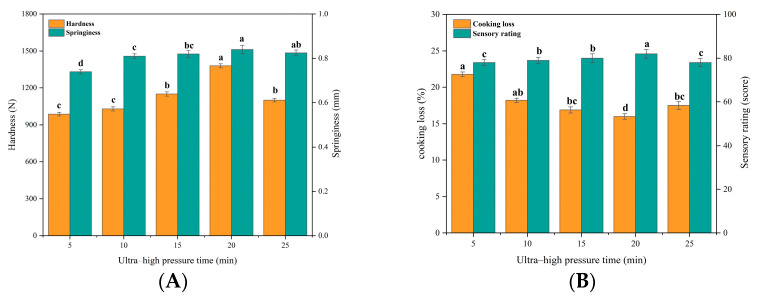
Effect of UHP time on texture, cooking loss, and sensory properties of beef patties. (**A**) Effect of UHP time (min) on hardness (N) and springiness (mm) of beef patties. (**B**) Effect of UHP time on cooking loss (%) and sensory characteristics (score) of beef patties. Different letters (a–d) indicate significant differences (*p* < 0.05).

**Figure 3 foods-12-03092-f003:**
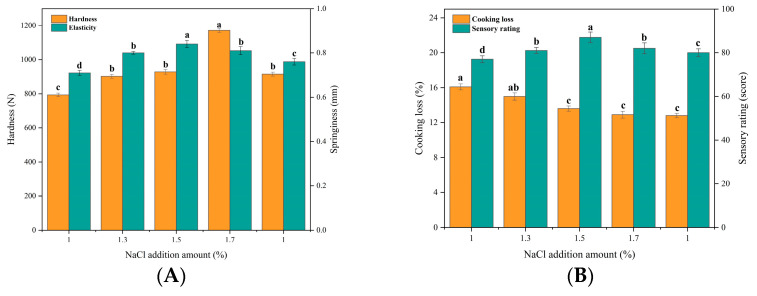
Effect of NaCl addition on texture, cooking loss, and sensory properties of beef patties. (**A**) Effect of Nacl addition (%) on hardness (N) and springiness (mm) of beef patties. (**B**) Effect of NaCl addition (%)on cooking loss (%) and sensory characteristics (score) of beef patties. Different letters (a–d) indicate significant differences (*p* < 0.05).

**Figure 4 foods-12-03092-f004:**
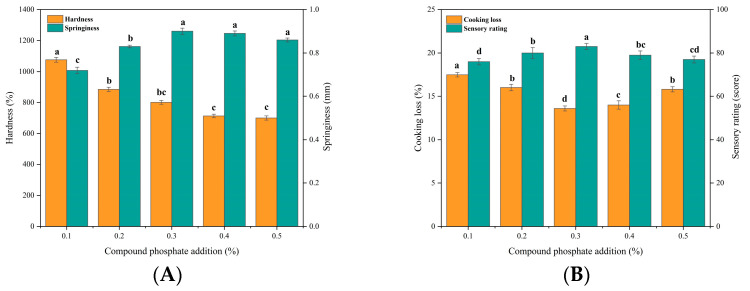
Effect of compound phosphate addition on texture, cooking loss and sensory properties of beef patties. (**A**) Effect of compound phosphate addition (%) on hardness (N) and springiness (mm) of beef patties. (**B**) Effect of compound phosphate addition (%) on cooking loss (%) and sensory characteristics (score) of beef patties. Different letters (a–d) indicate significant differences (*p* < 0.05).

**Figure 5 foods-12-03092-f005:**
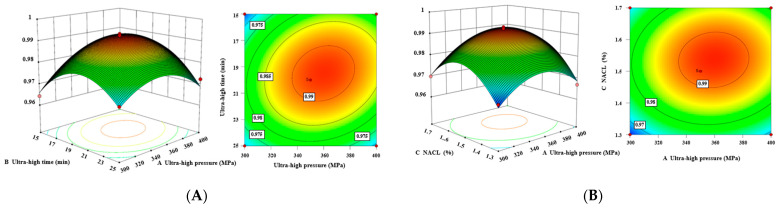
Impact of factor interactions on composite values. (**A**) Interaction of UHP pressure with UHP time. (**B**) Interaction of UHP pressure with NaCl addition. (**C**) Interaction of UHP pressure with compound phosphate addition. (**D**) Interaction between UHP time and NaCl addition. (**E**) Interaction of UHP time with the added amount of compound phosphate. (**F**) Interaction between the amount of NaCl added and the amount of compound phosphate added.

**Figure 6 foods-12-03092-f006:**
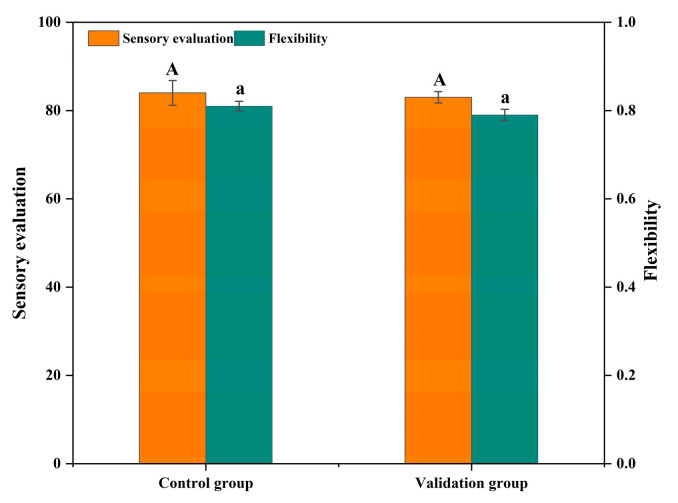
Validation experiment results. Upper case letters indicate significant differences in sensory scores and lower case letters indicate significant differences in springiness.

**Figure 7 foods-12-03092-f007:**
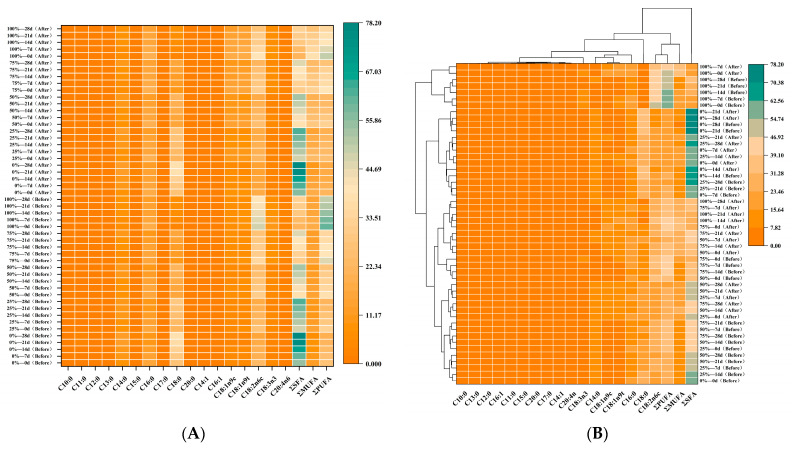
(**A**) Clustered heat map of fatty acid changes during refrigeration for different proportions of kraft gelatin-substituted fats. (**B**) Clustered heat map of fatty acid changes during refrigeration for kraft gelatin-substituted fats.

**Figure 8 foods-12-03092-f008:**
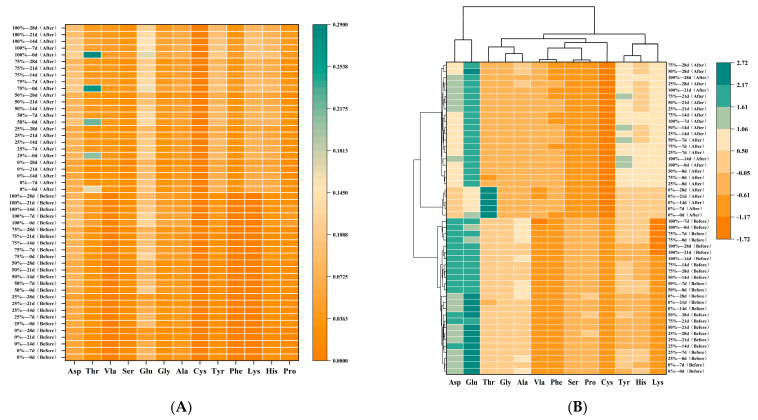
(**A**) Clustered heat map of amino acid changes during refrigeration for different proportions of kraft gelatin substituted fats. (**B**) Clustered heat map of amino acid changes during refrigeration for kraft gelatin substituted fats.

**Figure 9 foods-12-03092-f009:**
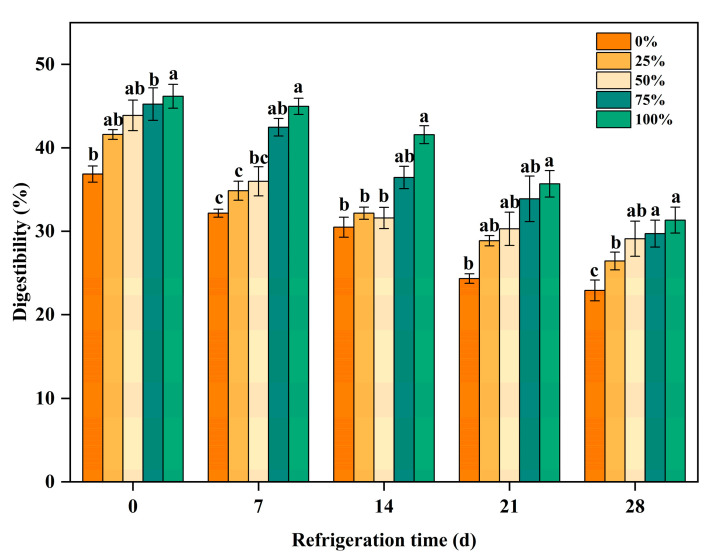
Changes in protein digestibility of beef patties in vitro during refrigeration. Lowercase letters indicate significant differences within the different treatment groups.

**Table 1 foods-12-03092-t001:** The formulation of beef patties made from cowhide gelatin.

Item (g)	0%	25%	50%	75%	100%
Beef	52	52	52	52	52
Beef fat	13	9.75	6.5	3.25	–
Cowhide gelatin	–	3.25	6.5	9.75	13
Salt	0.8	0.8	0.8	0.8	0.8
Pepper	0.1	0.1	0.1	0.1	0.1
Compound phosphate	0.15	0.15	0.15	0.15	0.15
Starch	13	13	13	13	13
Star anise	0.05	0.05	0.05	0.05	0.05
Clove	0.05	0.05	0.05	0.05	0.05
Fennel	0.05	0.05	0.05	0.05	0.05
Ice water	20.8	20.8	20.8	20.8	20.8

**Table 2 foods-12-03092-t002:** Response surface test factor level table.

Level	Factor			
	*A* Pressure/MPa	*B* Time/min	*C* NaCl/%	*D* Complex Phosphates/%
−1	300	15	1.3	0.2
0	350	20	1.5	0.3
1	400	25	1.7	0.4

**Table 3 foods-12-03092-t003:** Sensory scoring standards for beef patties.

Evaluation Projects	Evaluation Criteria	Score
Color (20)	Golden surface color, uniform	16–20
	Light yellow surface color, more uniform	8–15
	Surface color scorched black, uneven	1–7
Taste (20)	Pure meat flavor, with a rich meatloaf flavor	16–20
	Pure meat flavor, with a strong meatloaf flavor	8–15
	Meat flavor is lighter, meatloaf flavor is not strong	1–7
Taste (30)	Unique flavor, crisp, and juicy	25–30
	Average flavor, more crisp, less juicy	13–24
	Poor flavor, not crisp, basically no gravy	1–12
Organizational form (30)	Dense organization, not easy to loosen, good springiness	25–30
	Large tissue pores, easy to loosen, better springiness	13–24
	Loose tissue, loose, poor springiness	1–12

**Table 4 foods-12-03092-t004:** Experimental combinations of beef patties with the addition of kraft gelatin.

Run	Coded Values				Experimental Values		
	A	B	C	D	Y_1_	Y_2_	Y
1	−1	−1	0	0	0.82	80.95	0.97
2	1	−1	0	0	0.81	81.62	0.96
3	−1	1	0	0	0.81	80.46	0.96
4	1	1	0	0	0.84	80.33	0.97
5	0	0	−1	−1	0.82	81.18	0.96
6	0	0	1	−1	0.83	82.06	0.98
7	0	0	−1	1	0.81	81.90	0.96
8	0	0	1	1	0.82	81.57	0.97
9	−1	0	0	−1	0.81	81.91	0.97
10	1	0	0	−1	0.81	81.38	0.96
11	−1	0	0	1	0.81	80.34	0.96
12	1	0	0	1	0.81	82.80	0.98
13	0	−1	−1	0	0.82	81.23	0.97
14	0	1	−1	0	0.81	80.50	0.96
15	0	−1	1	0	0.82	81.58	0.97
16	0	1	1	0	0.83	81.33	0.97
17	−1	0	−1	0	0.82	81.42	0.97
18	1	0	−1	0	0.81	82.23	0.97
19	−1	0	1	0	0.84	80.00	0.97
20	1	0	1	0	0.81	82.98	0.97
21	0	−1	0	−1	0.81	81.20	0.96
22	0	1	0	−1	0.82	80.60	0.96
23	0	−1	0	1	0.81	80.82	0.96
24	0	1	0	1	0.82	80.80	0.96
25	0	0	0	0	0.81	84.23	0.98
26	0	0	0	0	0.83	83.09	0.98
27	0	0	0	0	0.82	83.29	0.98
28	0	0	0	0	0.82	84.50	0.99
29	0	0	0	0	0.83	83.34	0.98

Note: A means UHP; B means UHP time; C means NaCl addition; D means compound phosphate addition. Y_1_ denotes springiness (40%); Y_2_ denotes sensory score (60%); and Y denotes combined value.

**Table 5 foods-12-03092-t005:** Variance analysis of response surface tests.

Source of Variation	S.S.	DF.	M.S.	F-Value	*p*-Value	Sig.
Model	$2.429\times 10^{-3}$	14	$1.735\times 10^{-4}$	32.73	<0.0001	**
*A*	$1.202\times 10^{-4}$	1	$1.202\times 10^{-4}$	22.68	0.0003	**
*B*	$3.212\times 10^{-5}$	1	$3.212\times 10^{-5}$	6.06	0.0274	*
*C*	$1.919\times 10^{-4}$	1	$1.919\times 10^{-4}$	36.21	<0.0001	**
*D*	$3.585\times 10^{-7}$	1	$3.585\times 10^{-7}$	0.068	0.7986	
*AB*	$5.117\times 10^{-5}$	1	$5.117\times 10^{-5}$	9.65	0.0077	**
*AC*	$7.398\times 10^{-6}$	1	$7.398\times 10^{-6}$	1.40	0.2571	
*AD*	$6.369\times 10^{-5}$	1	$6.369\times 10^{-5}$	12.02	0.0038	**
*BC*	$2.128\times 10^{-5}$	1	$2.128\times 10^{-5}$	4.01	0.0649	
*BD*	$1.184\times 10^{-5}$	1	$1.184\times 10^{-5}$	2.23	0.1572	
*CD*	$1.243\times 10^{-7}$	1	$1.243\times 10^{-7}$	0.023	0.8805	
*A* ^2^	$5.327\times 10^{-4}$	1	$5.327\times 10^{-4}$	100.50	<0.0001	**
*B* ^2^	$8.725\times 10^{-4}$	1	$8.725\times 10^{-4}$	164.60	<0.0001	**
*C* ^2^	$8.979\times 10^{-4}$	1	$8.979\times 10^{-4}$	169.39	<0.0001	**
*D* ^2^	$6.915\times 10^{-4}$	1	$6.915\times 10^{-4}$	130.46	<0.0001	**
Residual	$7.421\times 10^{-5}$	14	$5.301\times 10^{-6}$			
Lack of fit	$6.320\times 10^{-5}$	10	$6.320\times 10^{-6}$	2.30	0.2197	
Net error	$1.101\times 10^{-5}$	4	$2.753\times 10^{-6}$			
Total deviation	$2.503\times 10^{-3}$	28				

Note: A means UHP; B means UHP time; C means NaCl addition; D means compound phosphate addition. S.S.: denotes sum. DF.: denotes degree of freedom. M.S.: denotes mean square. Sig.: indicates significance. “*” indicates significant difference (*p* < 0.05). “**” indicates highly significant difference (*p* < 0.01).

**Table 6 foods-12-03092-t006:** Effects of in vitro simulated digestion on fatty acid composition and content in fat replacement beef patties during refrigeration.

Items	R(%)	T										S.E		
		0 d		7 d		14 d		21 d		28 d			R	T
		Before	After	Before	After	Before	After	Before	After	Before	After			
C10:0	0	0.15 ± 0.01 ^Ba^	0.14 ± 0.01 ^Ba^	0.16 ± 0.01 ^ABa^	0.16 ± 0.01 ^ABa^	0.17 ± 0.01 ^ABa^	0.17 ± 0.01 ^ABa^	0.18 ± 0.01 ^ABa^	0.18 ± 0.01 ^Aa^	0.19 ± 0.01 ^Aa^	0.19 ± 0.01 ^Aa^	<0.01	**	**
	25	0.11 ± 0.01 ^Cb^	0.13 ± 0.01 ^Ba^	0.13 ± 0.01 ^BCb^	0.14 ± 0.01 ^Bb^	0.15 ± 0.01 ^ABab^	0.16 ± 0.01 ^ABa^	0.16 ± 0.01 ^ABa^	0.16 ± 0.01 ^ABb^	0.17 ± 0.01 ^Aa^	0.18 ± 0.01 ^Aab^			
	50	0.10 ± 0.01 ^Bb^	0.11 ± 0.01 ^Cb^	0.12 ± 0.01 ^ABb^	0.13 ± 0.01 ^BCb^	0.13 ± 0.01 ^ABbc^	0.14 ± 0.01 ^ABCb^	0.13 ± 0.01 ^ABb^	0.15 ± 0.01 ^ABbc^	0.14 ± 0.01 ^Ab^	0.17 ± 0.01 ^Abc^			
	75	0.07 ± 0.01 ^Cc^	0.09 ± 0.01 ^Cc^	0.08 ± 0.01 ^BCc^	0.11 ± 0.01 ^BCc^	0.11 ± 0.01 ^ABcd^	0.13 ± 0.01 ^ABb^	0.12 ± 0.01 ^Abc^	0.14 ± 0.01 ^ABcd^	0.13 ± 0.01 ^Abc^	0.16 ± 0.01 ^Ac^			
	100	0.05 ± 0.01 ^Cc^	0.07 ± 0.01 ^Cd^	0.07 ± 0.01 ^BCc^	0.09 ± 0.01 ^BCd^	0.09 ± 0.01 ^ABd^	0.11 ± 0.01 ^ABc^	0.10 ± 0.01 ^ABc^	0.13 ± 0.01 ^Ad^	0.12 ± 0.01 ^Ac^	0.14 ± 0.01 ^Ad^			
C11:0	0	1.50 ± 0.01 ^Da^	1.49 ± 0.01 ^Da^	1.68 ± 0.02 ^CDa^	1.65 ± 0.01 ^Ca^	1.81 ± 0.01 ^BCa^	1.78 ± 0.01 ^Ca^	1.98 ± 0.04 ^Ba^	1.97 ± 0.01 ^Ba^	2.23 ± 0.02 ^Aa^	2.17 ± 0.02 ^Aa^	<0.01	**	**
	25	1.31 ± 0.02 ^Cb^	1.29 ± 0.01 ^Db^	1.41 ± 0.03 ^Cb^	1.39 ± 0.04 ^Db^	1.63 ± 0.02 ^Bb^	1.58 ± 0.01 ^Cb^	1.77 ± 0.02 ^ABb^	1.74 ± 0.02 ^Bb^	1.90 ± 0.01 ^Ab^	1.86 ± 0.03 ^Ab^			
	50	1.04 ± 0.01 ^Dc^	1.02 ± 0.01 ^Ec^	1.20 ± 0.02 ^Cc^	1.16 ± 0.01 ^Dc^	1.41 ± 0.04 ^Bc^	1.38 ± 0.01 ^Cc^	1.56 ± 0.02 ^Bc^	1.54 ± 0.02 ^Bc^	1.74 ± 0.01 ^Ac^	1.70 ± 0.01 ^Ac^			
	75	0.76 ± 0.01 ^Dd^	0.72 ± 0.01 ^Dd^	0.82 ± 0.02 ^Dd^	0.80 ± 0.01 ^CDd^	0.96 ± 0.01 ^Cd^	0.93 ± 0.03 ^Cd^	1.11 ± 0.01 ^Bd^	1.10 ± 0.01 ^Bd^	1.33 ± 0.02 ^Ad^	1.30 ± 0.02 ^Ad^			
	100	0.50 ± 0.01 ^De^	0.47 ± 0.02 ^Ce^	0.65 ± 0.03 ^Ce^	0.60 ± 0.02 ^Be^	0.75 ± 0.03 ^BCe^	0.72 ± 0.01 ^Be^	0.87 ± 0.04 ^ABe^	0.85 ± 0.02 ^Ae^	0.96 ± 0.02 ^Ae^	0.93 ± 0.02 ^Ae^			
C12:0	0	0.18 ± 0.02 ^Da^	0.17 ± 0.01 ^Db^	0.22 ± 0.01 ^Ca^	0.20 ± 0.01 ^Db^	0.26 ± 0.01 ^Ba^	0.27 ± 0.02 ^Ca^	0.28 ± 0.02 ^Ba^	0.32 ± 0.01 ^Ba^	0.37 ± 0.02 ^Aa^	0.37 ± 0.03 ^Aa^	<0.01	**	**
	25	0.16 ± 0.02 ^Bab^	0.20 ± 0.01 ^Ca^	0.24 ± 0.01 ^Aa^	0.25 ± 0.01 ^Ba^	0.25 ± 0.01 ^Aa^	0.26 ± 0.02 ^Ba^	0.27 ± 0.02 ^Aa^	0.27 ± 0.03 ^ABb^	0.27 ± 0.04 ^Ab^	0.30 ± 0.01 ^Ab^			
	50	0.15 ± 0.01 ^Cab^	0.16 ± 0.01 ^Cbc^	0.18 ± 0.01 ^BCb^	0.17 ± 0.01 ^Cbc^	0.21 ± 0.02 ^Bb^	0.21 ± 0.01 ^Bb^	0.21 ± 0.01 ^Bb^	0.23 ± 0.01 ^Bc^	0.27 ± 0.02 ^Ab^	0.25 ± 0.01 ^Ac^			
	75	0.13 ± 0.02 ^Cbc^	0.13 ± 0.01 ^Dc^	0.15 ± 0.02 ^Cbc^	0.15 ± 0.01 ^CDcd^	0.16 ± 0.01 ^BCc^	0.17 ± 0.02 ^BCbc^	0.19 ± 0.02 ^ABb^	0.19 ± 0.01 ^ABd^	0.22 ± 0.02 ^Abc^	0.22 ± 0.01 ^Acd^			
	100	0.10 ± 0.02 ^Dc^	0.10 ± 0.01 ^Dd^	0.13 ± 0.02 ^CDc^	0.13 ± 0.01 ^CDd^	0.15 ± 0.02 ^BCc^	0.15 ± 0.01 ^BCc^	0.17 ± 0.03 ^ABb^	0.17 ± 0.01 ^ABd^	0.19 ± 0.02 ^Ac^	0.20 ± 0.02 ^Ad^			
C13:0	0	0.13 ± 0.01 ^Ca^	0.14 ± 0.01 ^Da^	0.16 ± 0.01 ^Ca^	0.17 ± 0.01 ^CDa^	0.20 ± 0.02 ^Ba^	0.20 ± 0.01 ^BCa^	0.23 ± 0.02 ^ABa^	0.23 ± 0.01 ^ABa^	0.26 ± 0.01 ^Aa^	0.25 ± 0.01 ^Aa^	<0.01	**	**
	25	0.12 ± 0.02 ^Da^	0.13 ± 0.02 ^Ba^	0.14 ± 0.02 ^CDab^	0.14 ± 0.01 ^Bb^	0.16 ± 0.02 ^BCab^	0.16 ± 0.01 ^Bb^	0.19 ± 0.02 ^ABab^	0.20 ± 0.02 ^Aa^	0.20 ± 0.01 ^Ab^	0.21 ± 0.01 ^Ab^			
	50	0.10 ± 0.02 ^Cab^	0.10 ± 0.01 ^Cab^	0.11 ± 0.03 ^Cab^	0.12 ± 0.01 ^BCbc^	0.13 ± 0.02 ^BCbc^	0.14 ± 0.01 ^Bc^	0.15 ± 0.02 ^ABbc^	0.15 ± 0.01 ^ABb^	0.18 ± 0.03 ^Ab^	0.18 ± 0.01 ^Ac^			
	75	0.08 ± 0.02 ^Cab^	0.08 ± 0.01 ^Db^	0.10 ± 0.03 ^Cab^	0.10 ± 0.01 ^CDcd^	0.11 ± 0.01 ^BCbc^	0.12 ± 0.01 ^BCd^	0.14 ± 0.02 ^ABbc^	0.14 ± 0.01 ^ABbc^	0.16 ± 0.02 ^Abc^	0.16 ± 0.01 ^Acd^			
	100	0.06 ± 0.02 ^Cb^	0.07 ± 0.02 ^Cb^	0.08 ± 0.02 ^BCb^	0.08 ± 0.01 ^BCd^	0.09 ± 0.03 ^BCc^	0.09 ± 0.01 ^BCe^	0.11 ± 0.03 ^ABc^	0.11 ± 0.01 ^ABc^	0.13 ± 0.02 ^Ac^	0.13 ± 0.01 ^Ad^			
C14:0	0	8.22 ± 0.08 ^Ea^	8.27 ± 0.08 ^Ea^	9.89 ± 0.06 ^Da^	9.80 ± 0.08 ^Da^	10.37 ± 0.07 ^Ca^	10.67 ± 0.07 ^Ca^	11.90 ± 0.09 ^Ba^	11.87 ± 0.06 ^Ba^	12.69 ± 0.09 ^Aa^	12.67 ± 0.02 ^Aa^	<0.01	**	**
	25	8.10 ± 0.01 ^Ea^	8.04 ± 0.03 ^Eb^	8.61 ± 0.09 ^Db^	8.70 ± 0.02 ^Db^	9.53 ± 0.02 ^Cb^	9.54 ± 0.04 ^Cb^	10.11 ± 0.03 ^Bb^	10.13 ± 0.03 ^Bb^	11.16 ± 0.05 ^Ab^	11.20 ± 0.02 ^Ab^			
	50	7.71 ± 0.04 ^Eb^	7.70 ± 0.01 ^Ec^	8.53 ± 0.06 ^Dc^	8.61 ± 0.02 ^Db^	8.88 ± 0.09 ^Cc^	8.94 ± 0.08 ^Cc^	9.79 ± 0.11 ^Bc^	9.80 ± 0.01 ^Bc^	10.94 ± 0.06 ^Ac^	10.89 ± 0.03 ^Ac^			
	75	6.79 ± 0.04 ^Ec^	6.80 ± 0.03 ^Ed^	7.17 ± 0.08 ^Dd^	7.22 ± 0.05 ^Dc^	7.75 ± 0.15 ^Cd^	7.81 ± 0.12 ^Cd^	8.96 ± 0.03 ^Bd^	8.91 ± 0.03 ^Bd^	9.90 ± 0.01 ^Ad^	9.93 ± 0.02 ^Ad^			
	100	5.74 ± 0.08 ^Ed^	5.78 ± 0.05 ^Ee^	6.15 ± 0.06 ^d^	6.19 ± 0.03 ^Dd^	7.50 ± 0.03 ^Cd^	7.49 ± 0.04 ^Ce^	7.71 ± 0.02 ^Be^	7.69 ± 0.09 ^Be^	8.45 ± 0.05 ^Ae^	8.47 ± 0.06 ^Ae^			
C15:0	0	1.29 ± 0.11 ^Da^	1.28 ± 0.07 ^Da^	1.59 ± 0.04 ^Ca^	1.62 ± 0.04 ^Ca^	1.84 ± 0.06 ^Ba^	1.88 ± 0.06 ^Ba^	1.90 ± 0.06 ^ABa^	1.91 ± 0.06 ^Ba^	2.04 ± 0.04 ^Aa^	2.09 ± 0.04 ^Aa^	<0.01	**	**
	25	1.04 ± 0.04 ^Cb^	1.06 ± 0.05 ^Db^	1.18 ± 0.09 ^Cb^	1.21 ± 0.06 ^CDb^	1.36 ± 0.09 ^Bb^	1.39 ± 0.04 ^BCb^	1.50 ± 0.05 ^ABb^	1.52 ± 0.06 ^ABb^	1.60 ± 0.06 ^Ab^	1.60 ± 0.11 ^Ab^			
	50	0.92 ± 0.04 ^Dc^	0.98 ± 0.01 ^Cb^	0.95 ± 0.06 ^CDc^	0.93 ± 0.03 ^BCc^	1.02 ± 0.01 ^BCc^	1.07 ± 0.02 ^BCc^	1.11 ± 0.02 ^Bc^	1.13 ± 0.02 ^Bc^	1.28 ± 0.04 ^Ac^	1.34 ± 0.04 ^Ac^			
	75	0.73 ± 0.05 ^Dd^	0.72 ± 0.11 ^Cc^	0.85 ± 0.05 ^CDc^	0.86 ± 0.04 ^BCc^	0.94 ± 0.03 ^BCd^	0.95 ± 0.06 ^ABc^	1.05 ± 0.04 ^ABc^	1.03 ± 0.01 ^Ac^	1.10 ± 0.01 ^Ad^	1.10 ± 0.03 ^Ad^			
	100	0.61 ± 0.02 ^De^	0.60 ± 0.04 ^Cc^	0.70 ± 0.02 ^CDd^	0.69 ± 0.05 ^BCd^	0.81 ± 0.03 ^BCe^	0.80 ± 0.07 ^ABCd^	0.91 ± 0.02 ^ABd^	0.89 ± 0.04 ^ABd^	1.00 ± 0.03 ^Ae^	0.99 ± 0.04 ^Ad^			
C16:0	0	15.71 ± 0.04 ^Aa^	15.91 ± 0.12 ^Aa^	16.22 ± 0.0 4 ^Ba^	16.30 ± 0.02 ^Ba^	17.30 ± 0.02 ^Ca^	17.31 ± 0.03 ^Ca^	18.75 ± 0.08 ^Da^	18.80 ± 0.09 ^Da^	19.04 ± 0.04 ^Ea^	19.08 ± 0.05 ^Ea^	<0.01	**	**
	25	13.82 ± 0.08 ^Ab^	13.90 ± 0.02 ^Ab^	15.59 ± 0.04 ^Bb^	15.68 ± 0.06 ^Bb^	15.89 ± 0.03 ^Cb^	15.93 ± 0.05 ^Cb^	16.62 ± 0.05 ^Db^	16.68 ± 0.06 ^Db^	17.51 ± 0.04 ^Eb^	17.58 ± 0.06 ^Eb^			
	50	12.29 ± 0.06 ^Ac^	12.41 ± 0.02 ^Ac^	13.06 ± 0.03 ^Bc^	13.14 ± 0.04 ^Bc^	14.42 ± 0.05 ^Cc^	14.51 ± 0.02 ^Cc^	15.44 ± 0.14 ^Dc^	15.50 ± 0.01 ^Dc^	16.22 ± 0.04 ^Ec^	16.30 ± 0.02 ^Ec^			
	75	10.74 ± 0.06 ^Ad^	10.78 ± 0.06 ^Ad^	11.73 ± 0.06 ^Bd^	11.67 ± 0.04 ^Bd^	13.31 ± 0.02 ^Cd^	13.30 ± 0.03 ^Cd^	14.11 ± 0.04 ^Dd^	14.20 ± 0.03 ^Dd^	15.12 ± 0.01 ^Ed^	15.17 ± 0.03 ^Ed^			
	100	9.95 ± 0.06 ^Ae^	10.08 ± 0.05 ^Ae^	10.50 ± 0.02 ^Be^	10.57 ± 0.07 ^Be^	11.44 ± 0.08 ^Ce^	11.46 ± 0.04 ^Ce^	12.74 ± 0.06 ^De^	12.77 ± 0.08 ^De^	13.47 ± 0.06 ^Ee^	13.43 ± 0.06 ^Ee^			
C17:0	0	0.87 ± 0.01 ^Aa^	0.84 ± 0.02 ^Aa^	0.95 ± 0.02 ^Ba^	0.96 ± 0.01 ^ABa^	1.07 ± 0.02 ^Ca^	1.09 ± 0.02 ^BCa^	1.11 ± 0.03 ^CDa^	1.13 ± 0.03 ^CDa^	1.17 ± 0.04 ^Da^	1.18 ± 0.04 ^Da^	<0.01	**	**
	25	0.75 ± 0.04 ^Ab^	0.80 ± 0.01 ^Aa^	0.82 ± 0.01 ^ABb^	0.84 ± 0.02 ^ABb^	0.94 ± 0.01 ^BCb^	0.93 ± 0.03 ^BCb^	0.98 ± 0.01 ^Cb^	0.99 ± 0.05 ^Cb^	1.04 ± 0.02 ^Db^	1.05 ± 0.01 ^Cb^			
	50	0.63 ± 0.02 ^Ac^	0.68 ± 0.02 ^Ab^	0.70 ± 0.02 ^ABc^	0.72 ± 0.01 ^ABc^	0.78 ± 0.02 ^BCc^	0.84 ± 0.02 ^BCc^	0.83 ± 0.02 ^CDc^	0.87 ± 0.01 ^BCc^	0.96 ± 0.02 ^Dc^	0.98 ± 0.01 ^Cc^			
	75	0.58 ± 0.02 ^Ad^	0.61 ± 0.02 ^Ac^	0.63 ± 0.02 ^ABd^	0.67 ± 0.01 ^Bc^	0.71 ± 0.02 ^Bd^	0.72 ± 0.01 ^Cd^	0.80 ± 0.02 ^Bc^	0.87 ± 0.02 ^Cc^	0.91 ± 0.02 ^Bcd^	0.97 ± 0.03 ^Dc^			
	100	0.47 ± 0.02 ^Ae^	0.47 ± 0.04 ^Ad^	0.50 ± 0.02 ^Ae^	0.52 ± 0.04 ^Bd^	0.63 ± 0.01 ^Be^	0.64 ± 0.01 ^BCe^	0.74 ± 0.01 ^Bd^	0.75 ± 0.01 ^BCd^	0.87 ± 0.02 ^Bd^	0.88 ± 0.02 ^Cd^			
C18:0	0	26.35 ± 0.06 ^Ea^	26.76 ± 0.18 ^Ea^	29.41 ± 0.04 ^Da^	29.50 ± 0.04 ^Da^	32.91 ± 0.09 ^Ca^	32.76 ± 0.15 ^Ca^	36.98 ± 0.06 ^Ba^	37.40 ± 0.23 ^Ba^	38.53 ± 0.05 ^Aa^	38.70 ± 0.13 ^Aa^	<0.01	**	**
	25	20.63 ± 0.05 ^Eb^	20.77 ± 0.17 ^Eb^	22.70 ± 0.07 ^Db^	22.60 ± 0.12 ^Db^	24.31 ± 0.04 ^Cb^	24.61 ± 0.11 ^Cb^	26.25 ± 0.08 ^Bb^	26.58 ± 0.29 ^Bb^	27.93 ± 0.06 ^Ab^	27.77 ± 0.11 ^Ab^			
	50	14.94 ± 0.04 ^Ec^	15.07 ± 0.06 ^Ec^	16.16 ± 0.06 ^Dc^	16.42 ± 0.09 ^Dc^	17.47 ± 0.07 ^Cc^	17.60 ± 0.03 ^Cc^	19.15 ± 0.05 ^Bc^	19.26 ± 0.07 ^Bc^	20.69 ± 0.06 ^Ac^	20.79 ± 0.06 ^Ac^			
	75	9.82 ± 0.04 ^Ed^	9.79 ± 0.17 ^Ed^	11.26 ± 0.04 ^Dd^	11.41 ± 0.04 ^Dd^	12.81 ± 0.04 ^Cd^	12.88 ± 0.05 ^Cd^	13.45 ± 0.06 ^Bd^	13.58 ± 0.06 ^Bd^	15.60 ± 0.03 ^Ad^	15.84 ± 0.08 ^Ad^			
	100	3.36 ± 0.05 ^Ce^	3.47 ± 0.28 ^Ee^	4.04 ± 0.04 ^BCe^	4.15 ± 0.05 ^De^	6.39 ± 0.05 ^ABe^	6.61 ± 0.12 ^Ce^	6.72 ± 0.11 ^Ae^	6.88 ± 0.06 ^Be^	7.34 ± 0.06 ^Ae^	7.46 ± 0.05 ^Ae^			
C20:0	0	1.05 ± 0.05 ^Ca^	1.08 ± 0.06 ^Ca^	1.16 ± 0.04 ^BCa^	1.19 ± 0.03 ^BCa^	1.23 ± 0.03 ^ABCa^	1.23 ± 0.06 ^BCa^	1.34 ± 0.04 ^ABa^	1.35 ± 0.04 ^ABa^	1.50 ± 0.06 ^Aa^	1.47 ± 0.04 ^Aa^	<0.01	**	**
	25	0.81 ± 0.03 ^Cb^	0.87 ± 0.02 ^Db^	0.92 ± 0.04 ^BCb^	0.94 ± 0.04 ^CDb^	1.01 ± 0.02 ^Bb^	1.03 ± 0.02 ^BCb^	1.09 ± 0.04 ^ABb^	1.15 ± 0.05 ^Bb^	1.25 ± 0.05 ^Ab^	1.29 ± 0.04 ^Ab^			
	50	0.61 ± 0.02 ^Cc^	0.64 ± 0.04 ^Cc^	0.67 ± 0.02 ^Cc^	0.70 ± 0.02 ^Cc^	0.84 ± 0.04 ^Bc^	0.87 ± 0.02 ^Bc^	0.90 ± 0.02 ^Bc^	0.93 ± 0.02 ^Bc^	1.05 ± 0.06 ^Ac^	1.12 ± 0.01 ^Ac^			
	75	0.41 ± 0.04 ^Bd^	0.44 ± 0.04 ^Bd^	0.45 ± 0.05 ^Bd^	0.49 ± 0.03 ^Bd^	0.55 ± 0.04 ^Bd^	0.60 ± 0.01 ^Bd^	0.75 ± 0.05 ^Ad^	0.81 ± 0.02 ^Ad^	0.81 ± 0.02 ^Ad^	0.81 ± 0.04 ^Ad^			
	100	0.11 ± 0.02 ^De^	0.12 ± 0.01 ^De^	0.30 ± 0.02 ^Ce^	0.35 ± 0.03 ^Ce^	0.43 ± 0.03 ^BCe^	0.46 ± 0.05 ^BCe^	0.49 ± 0.04 ^Be^	0.51 ± 0.03 ^Be^	0.62 ± 0.04 ^Ae^	0.67 ± 0.03 ^Ae^			
C14:1	0	1.33 ± 0.02 ^Ad^	1.35 ± 0.01 ^Ae^	1.23 ± 0.03 ^Ad^	1.26 ± 0.02 ^Ae^	0.94 ± 0.01 ^Bd^	1.06 ± 0.02 ^Be^	0.88 ± 0.03 ^Be^	0.98 ± 0.01 ^BCe^	0.71 ± 0.03 ^Ce^	0.87 ± 0.03 ^Ce^	<0.01	**	**
	25	1.37 ± 0.01 ^Ad^	1.45 ± 0.03 ^Ad^	1.29 ± 0.03 ^Ad^	1.35 ± 0.02 ^ABd^	1.18 ± 0.03 ^ABc^	1.23 ± 0.01 ^BCd^	1.06 ± 0.03 ^BCd^	1.07 ± 0.03 ^CDd^	0.93 ± 0.02 ^Cd^	0.97 ± 0.01 ^Dd^			
	50	1.46 ± 0.06 ^Ac^	1.57 ± 0.04 ^Ac^	1.36 ± 0.02 ^ABc^	1.45 ± 0.01 ^ABc^	1.24 ± 0.03 ^BCc^	1.36 ± 0.01 ^ABc^	1.13 ± 0.02 ^CDc^	1.24 ± 0.02 ^BCc^	1.01 ± 0.03 ^Dc^	1.08 ± 0.02 ^Cc^			
	75	1.58 ± 0.02 ^Ab^	1.67 ± 0.04 ^Ab^	1.47 ± 0.04 ^ABb^	1.57 ± 0.04 ^ABb^	1.35 ± 0.04 ^BCb^	1.42 ± 0.01 ^BCb^	1.27 ± 0.03 ^CDb^	1.36 ± 0.01 ^CDb^	1.16 ± 0.04 ^Db^	1.22 ± 0.01 ^Db^			
	100	1.66 ± 0.03 ^Aa^	1.79 ± 0.01 ^Aa^	1.59 ± 0.03 ^ABa^	1.64 ± 0.01 ^ABa^	1.49 ± 0.04 ^ABa^	1.53 ± 0.02 ^BCa^	1.40 ± 0.02 ^BCa^	1.48 ± 0.01 ^BCa^	1.24 ± 0.03 ^Ca^	1.37 ± 0.03 ^Ca^			
C16:1	0	0.12 ± 0.01 ^Ac^	0.20 ± 0.01 ^Ae^	0.11 ± 0.01 ^ABc^	0.18 ± 0.01 ^ABe^	0.08 ± 0.01 ^ABd^	0.14 ± 0.01 ^BCd^	0.07 ± 0.01 ^ABc^	0.11 ± 0.01 ^Ce^	0.05 ± 0.02 ^Bb^	0.10 ± 0.01 ^Cd^	<0.01	**	**
	25	0.14 ± 0.02 ^Ac^	0.30 ± 0.02 ^Ad^	0.12 ± 0.01 ^ABc^	0.24 ± 0.01 ^ABd^	0.12 ± 0.01 ^ABc^	0.20 ± 0.01 ^BCc^	0.09 ± 0.01 ^ABc^	0.17 ± 0.01 ^Cd^	0.08 ± 0.01 ^Bb^	0.15 ± 0.01 ^Cc^			
	50	0.23 ± 0.04 ^Ab^	0.35 ± 0.01 ^Ac^	0.20 ± 0.01 ^ABb^	0.30 ± 0.01 ^ABc^	0.18 ± 0.02 ^Bb^	0.27 ± 0.01 ^BCb^	0.16 ± 0.01 ^Cb^	0.23 ± 0.01 ^BCc^	0.13 ± 0.02 ^Ca^	0.20 ± 0.01 ^Cb^			
	75	0.27 ± 0.03 ^Aab^	0.40 ± 0.01 ^Ab^	0.23 ± 0.03 ^ABab^	0.35 ± 0.01 ^ABb^	0.20 ± 0.01 ^BCb^	0.31 ± 0.02 ^BCb^	0.17 ± 0.02 ^CDab^	0.27 ± 0.01 ^Cb^	0.13 ± 0.03 ^Da^	0.24 ± 0.02 ^Db^			
	100	0.31 ± 0.02 ^Aa^	0.45 ± 0.01 ^Aa^	0.27 ± 0.02 ^ABa^	0.41 ± 0.02 ^ABa^	0.25 ± 0.01 ^BCa^	0.37 ± 0.02 ^BCa^	0.20 ± 0.01 ^CDa^	0.33 ± 0.01 ^CDa^	0.17 ± 0.02 ^Da^	0.30 ± 0.01 ^Da^			
C18:1n9c	0	4.82 ± 0.04 ^Ae^	7.45 ± 0.10 ^Ae^	4.14 ± 0.04 ^Be^	6.97 ± 0.07 ^Be^	3.81 ± 0.06 ^Ce^	6.37 ± 0.30 ^Ce^	3.23 ± 0.06 ^De^	5.89 ± 0.06 ^De^	2.80 ± 0.06 ^Ee^	5.08 ± 0.08 ^Ee^	<0.01	**	**
	25	5.11 ± 0.03 ^Ad^	10.43 ± 0.08 ^Ad^	4.84 ± 0.07 ^Bd^	9.78 ± 0.06 ^Bd^	4.60 ± 0.01 ^Cd^	8.50 ± 0.12 ^Cd^	4.40 ± 0.03 ^Dd^	7.96 ± 0.05 ^Dd^	4.29 ± 0.05 ^Ed^	6.92 ± 0.04 ^Ed^			
	50	6.04 ± 0.04 ^Ac^	13.35 ± 0.08 ^Ac^	5.62 ± 0.04 ^Bc^	12.09 ± 0.11 ^Bc^	5.32 ± 0.04 ^Cc^	11.17 ± 0.11 ^Cc^	5.03 ± 0.02 ^Dc^	10.31 ± 0.06 ^Dc^	4.72 ± 0.04 ^Ec^	9.42 ± 0.08 ^Ec^			
	75	6.74 ± 0.04 ^Ab^	15.85 ± 0.10 ^Ab^	6.34 ± 0.06 ^Bb^	13.79 ± 0.08 ^Bb^	6.00 ± 0.03 ^Cb^	12.80 ± 0.06 ^Cb^	5.81 ± 0.04 ^Db^	11.61 ± 0.04 ^Db^	5.62 ± 0.04 ^Eb^	10.32 ± 0.09 ^Eb^			
	100	7.33 ± 0.05 ^Aa^	17.66 ± 0.10 ^Aa^	7.15 ± 0.06 ^Ba^	15.45 ± 0.33 ^Ba^	7.04 ± 0.04 ^Ca^	14.90 ± 0.04 ^Ca^	6.84 ± 0.04 ^Da^	13.59 ± 0.09 ^Da^	6.34 ± 0.06 ^Ea^	12.25 ± 0.10 ^Ea^			
C18:1n9t	0	5.31 ± 0.13 ^Ae^	8.37 ± 0.13 ^Ae^	4.63 ± 0.05 ^Be^	7.32 ± 0.05 ^Be^	4.07 ± 0.08 ^Ce^	6.38 ± 0.29 ^Ce^	3.72 ± 0.04 ^De^	5.35 ± 0.09 ^De^	3.53 ± 0.06 ^Ed^	4.24 ± 0.10 ^Ee^	<0.01	**	**
	25	6.72 ± 0.09 ^Ad^	10.25 ± 0.10 ^Ad^	5.48 ± 0.06 ^Bd^	9.44 ± 0.14 ^Bd^	5.02 ± 0.04 ^Cd^	8.35 ± 0.12 ^Cd^	4.81 ± 0.03 ^Dd^	7.81 ± 0.12 ^Dd^	4.68 ± 0.04 ^Ec^	6.80 ± 0.08 ^Ed^			
	50	7.14 ± 0.04 ^Ac^	13.34 ± 0.12 ^Ac^	6.44 ± 0.06 ^Bc^	11.78 ± 0.16 ^Bc^	5.79 ± 0.04 ^Cc^	10.81 ± 0.04 ^Cc^	5.13 ± 0.02 ^Dc^	9.61 ± 0.04 ^Dc^	4.89 ± 0.04 ^Eb^	8.43 ± 0.05 ^Ec^			
	75	7.94 ± 0.07 ^Ab^	15.54 ± 0.08 ^Ab^	7.26 ± 0.09 ^Bb^	14.34 ± 0.08 ^Bb^	6.52 ± 0.09 ^Cb^	13.83 ± 0.09 ^Cb^	6.06 ± 0.07 ^Db^	12.59 ± 0.09 ^Db^	5.53 ± 0.05 ^Eb^	11.12 ± 0.05 ^Eb^			
	100	8.61 ± 0.07 ^Aa^	17.91 ± 0.06 ^Aa^	8.14 ± 0.04 ^Ba^	16.80 ± 0.05 ^Ba^	7.63 ± 0.01 ^Ca^	15.68 ± 0.08 ^Ca^	7.07 ± 0.02 ^Da^	14.78 ± 0.13 ^Da^	6.90 ± 0.01 ^Ea^	13.83 ± 0.06 ^Ea^			
C18:2n6c	0	22.63 ± 0.06 ^Ae^	18.28 ± 0.06 ^Ae^	20.14 ± 0.04 ^Be^	16.30 ± 0.05 ^Be^	19.86 ± 0.05 ^Cd^	15.35 ± 0.05 ^Ce^	17.24 ± 0.04 ^De^	14.84 ± 0.04 ^De^	15.13 ± 0.06 ^Ee^	14.70 ± 0.03 ^Ee^	<0.01	**	**
	25	26.84 ± 0.04 ^Ad^	21.18 ± 0.08 ^Ad^	23.68 ± 0.11 ^Bd^	19.32 ± 0.04 ^Bd^	21.28 ± 0.23 ^Cc^	18.83 ± 0.06 ^Cd^	19.23 ± 0.07 ^Dd^	17.64 ± 0.08 ^Dd^	17.15 ± 0.13 ^Ed^	16.90 ± 0.04 ^Ed^			
	50	31.45 ± 0.09 ^Ac^	25.68 ± 0.21 ^Ac^	28.39 ± 0.05 ^Bc^	23.40 ± 0.06 ^Bc^	26.55 ± 0.18 ^Cc^	20.68 ± 0.06 ^Cc^	25.43 ± 0.06 ^Dc^	19.31 ± 0.05 ^Dc^	22.73 ± 0.05 ^Ec^	18.24 ± 0.04 ^Ec^			
	75	34.87 ± 0.08 ^Ab^	30.51 ± 0.04 ^Ab^	31.30 ± 0.03 ^Bb^	28.24 ± 0.06 ^Bb^	29.82 ± 0.04 ^Cb^	26.48 ± 0.08 ^Cb^	28.75 ± 0.06 ^Db^	24.74 ± 0.18 ^Db^	27.49 ± 0.04 ^Eb^	21.49 ± 0.06 ^Eb^			
	100	48.13 ± 0.06 ^Aa^	39.74 ± 0.06 ^Aa^	46.14 ± 0.04 ^Ba^	35.25 ± 0.10 ^Ba^	44.44 ± 0.06 ^Ca^	31.41 ± 0.06 ^Ca^	42.07 ± 0.06 ^Da^	29.82 ± 0.05 ^Da^	40.43 ± 0.16 ^Ea^	27.91 ± 0.11 ^Ea^			
C18:3n3	0	3.73 ± 0.05 ^Ae^	2.99 ± 0.04 ^Ae^	3.46 ± 0.06 ^Be^	2.81 ± 0.03 ^Ae^	3.33 ± 0.05 ^Bd^	2.54 ± 0.04 ^Be^	3.15 ± 0.05 ^Ce^	2.30 ± 0.01 ^BCe^	2.93 ± 0.06 ^De^	2.10 ± 0.02 ^Ce^	<0.01	**	**
	25	4.84 ± 0.08 ^Ad^	3.49 ± 0.06 ^Ad^	4.26 ± 0.04 ^Bd^	3.10 ± 0.02 ^Bd^	3.85 ± 0.06 ^Cc^	2.83 ± 0.09 ^Bd^	3.35 ± 0.04 ^Dd^	2.48 ± 0.04 ^Cd^	3.11 ± 0.03 ^Dd^	2.39 ± 0.05 ^Cd^			
	50	5.21 ± 0.02 ^Ac^	4.23 ± 0.08 ^Ac^	4.51 ± 0.02 ^Bc^	4.05 ± 0.05 ^Bc^	3.98 ± 0.06 ^Cc^	3.61 ± 0.12 ^Cc^	3.78 ± 0.13 ^Cc^	3.10 ± 0.02 ^Dc^	3.69 ± 0.04 ^Cc^	2.95 ± 0.05 ^Dc^			
	75	8.81 ± 0.02 ^Ab^	6.80 ± 0.05 ^Ab^	7.63 ± 0.05 ^Bb^	6.07 ± 0.06 ^Bb^	7.42 ± 0.06 ^Bb^	5.83 ± 0.06 ^BCb^	6.16 ± 0.04 ^Cb^	5.60 ± 0.04 ^Cb^	5.61 ± 0.03 ^Db^	5.00 ± 0.02 ^Db^			
	100	10.61 ± 0.13 ^Aa^	8.08 ± 0.06 ^Aa^	9.39 ± 0.06 ^Ba^	7.80 ± 0.05 ^Aa^	8.48 ± 0.1 ^Ca^	6.39 ± 0.04 ^Ba^	7.79 ± 0.05 ^Da^	6.09 ± 0.03 ^Ba^	7.07 ± 0.08 ^Ea^	5.72 ± 0.04 ^Ca^			
C20:4n6	0	1.24 ± 0.04 ^Ad^	1.05 ± 0.05 ^Ad^	1.12 ± 0.04 ^ABe^	0.86 ± 0.05 ^Be^	0.95 ± 0.02 ^BCe^	0.73 ± 0.02 ^BCe^	0.71 ± 0.08 ^CDe^	0.62 ± 0.01 ^CDd^	0.61 ± 0.07 ^De^	0.55 ± 0.05 ^Dd^	<0.01	**	**
	25	1.75 ± 0.05 ^Ac^	1.34 ± 0.06 ^Ac^	1.56 ± 0.05 ^Bd^	1.05 ± 0.05 ^Bd^	1.31 ± 0.04 ^Cd^	0.96 ± 0.05 ^BCd^	1.16 ± 0.04 ^CDd^	0.85 ± 0.03 ^CDc^	1.00 ± 0.03 ^Dd^	0.75 ± 0.04 ^Dc^			
	50	2.14 ± 0.04 ^Ab^	1.90 ± 0.04 ^Ab^	1.87 ± 0.03 ^Bc^	1.56 ± 0.04 ^Bc^	1.63 ± 0.02 ^BCc^	1.46 ± 0.04 ^BCc^	1.40 ± 0.03 ^CDc^	1.34 ± 0.04 ^Cb^	1.20 ± 0.02 ^Dc^	1.26 ± 0.04 ^Cb^			
	75	2.32 ± 0.04 ^Ab^	2.04 ± 0.01 ^Aa^	2.04 ± 0.04 ^Bb^	1.81 ± 0.11 ^Bb^	1.93 ± 0.04 ^Bb^	1.61 ± 0.04 ^Cb^	1.65 ± 0.05 ^Cb^	1.41 ± 0.03 ^Db^	1.42 ± 0.04 ^Db^	1.21 ± 0.02 ^Eb^			
	100	2.44 ± 0.04 ^Aa^	2.16 ± 0.05 ^Aa^	2.31 ± 0.02 ^Aa^	2.00 ± 0.02 ^Ba^	2.20 ± 0.02 ^ABa^	1.85 ± 0.04 ^Ca^	1.95 ± 0.06 ^BCa^	1.78 ± 0.01 ^Da^	1.69 ± 0.04 ^Ca^	1.61 ± 0.02 ^Ea^			
ΣSFA	0	55.42 ± 0.40 ^Aa^	56.04 ± 0.57 ^Aa^	61.42 ± 0.31 ^Ba^	61.52 ± 0.25 ^Ba^	67.13 ± 0.35 ^Ca^	67.34 ± 0.43 ^Ca^	74.63 ± 0.45 ^Da^	75.13 ± 0.55 ^Da^	77.99 ± 0.37 ^Ea^	78.14 ± 0.38 ^Ea^	<0.01	**	**
	25	46.82 ± 0.33 ^Ab^	47.17 ± 0.36 ^Ab^	51.71 ± 0.40 ^Bb^	51.85 ± 0.38 ^Bb^	55.20 ± 0.28 ^Cb^	55.55 ± 0.33 ^Cb^	58.91 ± 0.33 ^Db^	59.39 ± 0.62 ^Db^	63.00 ± 0.34 ^Eb^	63.01 ± 0.41 ^Eb^			
	50	38.46 ± 0.33 ^Ac^	38.83 ± 0.20 ^Ac^	41.65 ± 0.33 ^Bc^	42.07 ± 0.25 ^Bc^	45.26 ± 0.40 ^Cc^	45.66 ± 0.23 ^Cc^	49.24 ± 0.45 ^Dc^	49.53 ± 0.20 ^Dc^	53.45 ± 0.40 ^Ec^	53.69 ± 0.21 ^Ec^			
	75	30.07 ± 0.31 ^Ad^	30.14 ± 0.46 ^Ad^	33.20 ± 0.38 ^Bd^	33.46 ± 0.25 ^Bd^	37.38 ± 0.34 ^Cd^	37.59 ± 0.35 ^Cd^	40.65 ± 0.29 ^Dd^	40.94 ± 0.21 ^Dd^	45.25 ± 0.18 ^Ed^	45.63 ± 0.26 ^Ed^			
	100	20.91 ± 0.31 ^Ae^	21.20 ± 0.53 ^Ae^	23.09 ± 0.26 ^Be^	23.35 ± 0.32 ^Be^	28.27 ± 0.32 ^Ce^	28.50 ± 0.36 ^Ce^	30.53 ± 0.36 ^De^	30.72 ± 0.37 ^De^	33.11 ± 0.33 ^Ee^	33.26 ± 0.33 ^Ee^			
ΣMUFA	0	11.57 ± 0.21 ^Ae^	17.37 ± 0.25 ^Ae^	10.10 ± 0.13 ^Be^	15.72 ± 0.15 ^Be^	8.89 ± 0.17 ^Ce^	13.94 ± 0.63 ^Ce^	7.89 ± 0.13 ^De^	12.33 ± 0.18 ^De^	7.09 ± 0.15 ^Ee^	10.29 ± 0.23 ^Ee^	<0.01	**	**
	25	13.33 ± 0.16 ^Ad^	22.42 ± 0.23 ^Ad^	11.73 ± 0.18 ^Bd^	20.80 ± 0.24 ^Bd^	10.91 ± 0.08 ^Cd^	18.27 ± 0.27 ^Cd^	10.36 ± 0.10 ^Dd^	17.00 ± 0.21 ^Dd^	9.97 ± 0.12 ^Ed^	14.83 ± 0.14 ^Ed^			
	50	14.87 ± 0.18 ^Ac^	28.60 ± 0.25 ^Ac^	13.61 ± 0.13 ^Bc^	25.62 ± 0.30 ^Bc^	12.52 ± 0.13 ^Cc^	23.61 ± 0.18 ^Cc^	11.44 ± 0.08 ^Dc^	21.38 ± 0.13 ^Dc^	10.74 ± 0.13 ^Ec^	19.12 ± 0.16 ^Ec^			
	75	16.27 ± 0.16 ^Ab^	33.14 ± 0.23 ^Ab^	15.05 ± 0.22 ^Bb^	29.67 ± 0.21 ^Bb^	13.66 ± 0.18 ^Cb^	27.99 ± 0.18 ^Cb^	12.91 ± 0.16 ^Db^	25.44 ± 0.16 ^Db^	11.98 ± 0.15 ^Eb^	22.54 ± 0.18 ^Eb^			
	100	17.90 ± 0.17 ^Aa^	37.81 ± 0.18 ^Aa^	17.14 ± 0.14 ^Ba^	34.29 ± 0.40 ^Ba^	16.41 ± 0.11 ^Ca^	32.47 ± 0.16 ^Ca^	15.50 ± 0.10 ^Da^	30.17 ± 0.25 ^Da^	14.64 ± 0.13 ^Ea^	27.75 ± 0.20 ^Ea^			
ΣPUFA	0	27.59 ± 0.16 ^Ae^	22.31 ± 0.14 ^Ae^	24.72 ± 0.15 ^Be^	19.96 ± 0.13 ^Be^	24.13 ± 0.12 ^Ce^	18.61 ± 0.11 ^Ce^	21.09 ± 0.17 ^De^	17.76 ± 0.07 ^De^	18.67 ± 0.19 ^Ee^	17.34 ± 0.10 ^Ee^	<0.01	**	**
	25	33.42 ± 0.16 ^Ad^	26.00 ± 0.20 ^Ad^	29.49 ± 0.20 ^Bd^	23.46 ± 0.11 ^Bd^	26.43 ± 0.33 ^Cd^	22.61 ± 0.20 ^Cd^	23.74 ± 0.15 ^Dd^	20.97 ± 0.15 ^Dd^	21.26 ± 0.19 ^Ed^	20.04 ± 0.13 ^Ed^			
	50	38.79 ± 0.15 ^Ac^	31.80 ± 0.33 ^Ac^	34.76 ± 0.10 ^Bc^	29.01 ± 0.15 ^Bc^	32.15 ± 0.25 ^Cc^	25.74 ± 0.21 ^Cc^	30.60 ± 0.23 ^Dc^	23.74 ± 0.11 ^Dc^	27.61 ± 0.11 ^Ec^	22.44 ± 0.13 ^Ec^			
	75	45.99 ± 0.14 ^Ab^	39.34 ± 0.10 ^Ab^	40.96 ± 0.11 ^Bb^	36.12 ± 0.23 ^Bb^	39.17 ± 0.13 ^Cb^	33.92 ± 0.18 ^Cb^	36.56 ± 0.15 ^Db^	31.75 ± 0.25 ^Db^	34.52 ± 0.11 ^Eb^	27.69 ± 0.11 ^Eb^			
	100	61.17 ± 0.24 ^Aa^	49.97 ± 0.17 ^Aa^	57.83 ± 0.13 ^Ba^	45.04 ± 0.17 ^Ba^	55.11 ± 0.18 ^Ca^	39.65 ± 0.15 ^Ca^	51.80 ± 0.17 ^Da^	37.69 ± 0.09 ^Da^	49.19 ± 0.25 ^Ea^	35.23 ± 0.17 ^Ea^			

Note: Data are expressed as mean ± standard deviation. Upper case letters indicate significant differences in storage times. Lower case letters indicate significant differences in fat substitution rates. S.E: Standard error. n.s: Not significant. R: substitution rate. T: Cooling time. I. Interaction. ** *p* < 0.01.

**Table 7 foods-12-03092-t007:** Effects of simulated digestion in vitro on amino acid composition and content of beef patty during cold storage.

Items	R(%)	T										S.E	Sig.	
	0 d			7 d		14 d		21 d		28 d			R	T
		Before	After	Before	After	Before	After	Before	After	Before	After			
Asp	0	0.0582 ± 0.0001 ^Ae^	0.0707 ± 0.0001 ^Ae^	0.0574 ± 0.0001 ^Ae^	0.0672 ± 0.0002 ^Be^	0.0573 ± 0.0002 ^Ae^	0.0607 ± 0.0032 ^Ce^	0.0545 ± 0.0001 ^Be^	0.0593 ± 0.0001 ^Ce^	0.0504 ± 0.0001 ^Ce^	0.0565 ± 0.0001 ^De^	<0.01	**	**
	25	0.0648 ± 0.0001 ^Ad^	0.0835 ± 0.0001 ^Ad^	0.0652 ± 0.0009 ^Ad^	0.0753 ± 0.0002 ^Bd^	0.0614 ± 0.0001 ^Bd^	0.0727 ± 0.0002 ^Cd^	0.0587 ± 0.0011 ^Cd^	0.0665 ± 0.0002 ^Dd^	0.0494 ± 0.0001 ^Dd^	0.0635 ± 0.0001 ^Ed^			
	50	0.0702 ± 0.0001 ^Ac^	0.0974 ± 0.0001 ^Ac^	0.0772 ± 0.0001 ^Bc^	0.0893 ± 0.0002 ^Bc^	0.0752 ± 0.0004 ^Cc^	0.0830 ± 0.0002 ^Cc^	0.0698 ± 0.0009 ^Dc^	0.0757 ± 0.0001 ^Dc^	0.0650 ± 0.0001 ^Ec^	0.0730 ± 0.0001 ^Ec^			
	75	0.0821 ± 0.0002 ^Ab^	0.1131 ± 0.0001 ^Ab^	0.0803 ± 0.0002 ^Bb^	0.1073 ± 0.0001 ^Bb^	0.0771 ± 0.0001 ^Cb^	0.0983 ± 0.0002 ^Cb^	0.0729 ± 0.0001 ^Db^	0.0941 ± 0.0002 ^Db^	0.0687 ± 0.0002 ^Eb^	0.0835 ± 0.0003 ^Eb^			
	100	0.0974 ± 0.0001 ^Aa^	0.1156 ± 0.0002 ^Aa^	0.0942 ± 0.0008 ^Ba^	0.1115 ± 0.0001 ^Ba^	0.0903 ± 0.0002 ^Ca^	0.1032 ± 0.0002 ^Ca^	0.0874 ± 0.0012 ^Da^	0.0961 ± 0.0001 ^Da^	0.0828 ± 0.0003 ^Ea^	0.0937 ± 0.001 ^Ea^			
Thr	0	0.0280 ± 0.0001 ^Ae^	0.1681 ± 0.1921 ^Ae^	0.0262 ± 0.0001 ^Be^	0.0313 ± 0.0002 ^Be^	0.0261 ± 0.0001 ^Be^	0.0295 ± 0.0002 ^Ce^	0.0261 ± 0.0008 ^Bd^	0.0259 ± 0.0001 ^De^	0.0219 ± 0.0001 ^Ce^	0.0255 ± 0.0002 ^De^	<0.01	**	**
	25	0.0311 ± 0.0001 ^Ad^	0.2060 ± 0.2353 ^Ad^	0.0283 ± 0.0001 ^Bd^	0.0394 ± 0.0001 ^Bd^	0.0277 ± 0.0001 ^Bd^	0.0384 ± 0.0001 ^Bd^	0.0274 ± 0.0004 ^Bc^	0.0354 ± 0.0001 ^Cd^	0.0238 ± 0.0003 ^Cd^	0.0295 ± 0.0002 ^Dd^			
	50	0.0335 ± 0.0001 ^Ac^	0.2206 ± 0.2519 ^Ac^	0.0320 ± 0.0001 ^ABc^	0.0421 ± 0.0002 ^Bc^	0.0319 ± 0.0001 ^ABc^	0.0405 ± 0.0002 ^Cc^	0.0307 ± 0.0001 ^Bb^	0.0387 ± 0.0001 ^Dc^	0.0287 ± 0.0004 ^Cc^	0.0305 ± 0.0001 ^Ec^			
	75	0.0346 ± 0.0002 ^Ab^	0.2753 ± 0.3143 ^Ab^	0.0346 ± 0.0002 ^Ab^	0.0475 ± 0.0001 ^Bb^	0.0328 ± 0.0001 ^Bb^	0.0415 ± 0.0002 ^Cb^	0.0314 ± 0.0001 ^Bb^	0.0401 ± 0.0001 ^Db^	0.0299 ± 0.0001 ^Cb^	0.0371 ± 0.0002 ^Eb^			
	100	0.0520 ± 0.0001 ^Aa^	0.2896 ± 0.3307 ^Aa^	0.0421 ± 0.0004 ^Ba^	0.0521 ± 0.0001 ^Ba^	0.0414 ± 0.0001 ^BCa^	0.0503 ± 0.0001 ^Ba^	0.0401 ± 0.0002 ^Ca^	0.0481 ± 0.0001 ^Ca^	0.0376 ± 0.0001 ^Da^	0.0409 ± 0.0006 ^Da^			
Vla	0	0.0053 ± 0.0001 ^Ae^	0.0360 ± 0.0001 ^Aa^	0.0034 ± 0.0001 ^Be^	0.0341 ± 0.0001 ^Be^	0.0024 ± 0.0001 ^BCe^	0.0322 ± 0.0001 ^Ce^	0.0018 ± 0.0001 ^Ce^	0.0298 ± 0.0001 ^De^	0.0013 ± 0.0001 ^Ce^	0.0282 ± 0.0001 ^Ee^	<0.01	**	**
	25	0.0058 ± 0.0001 ^Ad^	0.0383 ± 0.0001 ^Aa^	0.0046 ± 0.0001 ^Bd^	0.0366 ± 0.0002 ^Bd^	0.0029 ± 0.0001 ^Cd^	0.0346 ± 0.001 ^Cd^	0.0027 ± 0.0001 ^Cd^	0.0323 ± 0.0003 ^Dd^	0.0016 ± 0.0001 ^Dd^	0.0309 ± 0.0001 ^Dd^			
	50	0.0074 ± 0.0001 ^Ac^	0.0414 ± 0.0001 ^Aa^	0.0059 ± 0.0001 ^Bc^	0.0398 ± 0.0001 ^Bc^	0.0044 ± 0.0001 ^Cc^	0.0378 ± 0.0001 ^Cc^	0.0035 ± 0.0001 ^CDc^	0.0341 ± 0.0001 ^Dc^	0.0022 ± 0.0001 ^Dc^	0.0326 ± 0.0001 ^Ec^			
	75	0.0088 ± 0.0001 ^Ab^	0.0385 ± 0.0099 ^Aa^	0.0069 ± 0.0001 ^Bb^	0.0425 ± 0.0001 ^Bb^	0.0059 ± 0.0001 ^BCb^	0.0396 ± 0.0001 ^Bb^	0.0049 ± 0.0001 ^Cb^	0.0386 ± 0.0001 ^Bb^	0.0035 ± 0.0001 ^Db^	0.0362 ± 0.0008 ^Cb^			
	100	0.0101 ± 0.0001 ^Aa^	0.0468 ± 0.0001 ^Aa^	0.0089 ± 0.0002 ^Ba^	0.0446 ± 0.0001 ^Ba^	0.0077 ± 0.0001 ^Ca^	0.0427 ± 0.0001 ^Ca^	0.0063 ± 0.0001 ^Da^	0.0420 ± 0.0005 ^Ca^	0.0053 ± 0.0001 ^Da^	0.0390 ± 0.0002 ^Da^			
Ser	0	0.0183 ± 0.0001 ^Be^	0.0269 ± 0.0001 ^Ae^	0.0199 ± 0.0001 ^Ae^	0.0242 ± 0.0001 ^Be^	0.0191 ± 0.0001 ^ABe^	0.0238 ± 0.0001 ^Be^	0.0163 ± 0.0001 ^Ce^	0.0223 ± 0.0001 ^Ce^	0.0139 ± 0.0001 ^De^	0.0178 ± 0.0001 ^De^	<0.01	**	**
	25	0.0234 ± 0.0001 ^Ad^	0.0282 ± 0.0001 ^Ad^	0.0224 ± 0.0001 ^Ad^	0.0273 ± 0.0001 ^ABd^	0.0200 ± 0.0001 ^Bd^	0.0268 ± 0.0002 ^Bd^	0.0182 ± 0.0002 ^Cd^	0.0247 ± 0.0001 ^Cd^	0.0162 ± 0.0004 ^Dd^	0.0188 ± 0.0001 ^Dd^			
	50	0.0261 ± 0.0001 ^Ac^	0.0303 ± 0.0001 ^Ac^	0.0254 ± 0.0003 ^Ac^	0.0302 ± 0.0002 ^Ac^	0.0241 ± 0.0001 ^Bc^	0.0292 ± 0.0002 ^Bc^	0.0238 ± 0.0001 ^Bc^	0.0281 ± 0.0001 ^Cc^	0.0182 ± 0.0001 ^Cc^	0.0191 ± 0.0001 ^Dc^			
	75	0.0265 ± 0.0001 ^Bb^	0.0404 ± 0.0001 ^Ab^	0.0304 ± 0.0001 ^Ab^	0.0332 ± 0.0001 ^Bb^	0.0258 ± 0.0001 ^BCb^	0.0306 ± 0.0002 ^Cb^	0.0246 ± 0.0001 ^Cb^	0.0296 ± 0.0001 ^Cb^	0.0215 ± 0.0002 ^Db^	0.0237 ± 0.0001 ^Db^			
	100	0.0393 ± 0.0001 ^Aa^	0.0437 ± 0.0001 ^Aa^	0.0356 ± 0.0002 ^Ba^	0.0368 ± 0.0001 ^Ba^	0.0297 ± 0.0001 ^Ca^	0.0343 ± 0.0002 ^Ca^	0.0281 ± 0.0001 ^Da^	0.0318 ± 0.0001 ^Da^	0.0254 ± 0.0001 ^Ea^	0.0282 ± 0.0002 ^Ea^			
Glu	0	0.0829 ± 0.0001 ^Ae^	0.1043 ± 0.0001 ^Ae^	0.0795 ± 0.0003 ^Be^	0.1041 ± 0.0001 ^Ae^	0.0668 ± 0.0003 ^Ce^	0.0956 ± 0.0002 ^Be^	0.0468 ± 0.0001 ^De^	0.0937 ± 0.0001 ^Ce^	0.0406 ± 0.0004 ^Ee^	0.0787 ± 0.0001 ^De^	<0.01	**	**
	25	0.0909 ± 0.0001 ^Ad^	0.1160 ± 0.0001 ^Ad^	0.0862 ± 0.0001 ^Bd^	0.1127 ± 0.0002 ^Bd^	0.0708 ± 0.0011 ^Cd^	0.1093 ± 0.0002 ^Cd^	0.0558 ± 0.0001 ^Dd^	0.0977 ± 0.0002 ^Dd^	0.0473 ± 0.0003 ^Ed^	0.0858 ± 0.0002 ^Ed^			
	50	0.1107 ± 0.0001 ^Ac^	0.1293 ± 0.0001 ^Ac^	0.0962 ± 0.0001 ^Bc^	0.1289 ± 0.0002 ^Ac^	0.0820 ± 0.0001 ^Cc^	0.1151 ± 0.0003 ^Bc^	0.0731 ± 0.0001 ^Dc^	0.1026 ± 0.0002 ^Cc^	0.0685 ± 0.0003 ^Ec^	0.0892 ± 0.0002 ^Dc^			
	75	0.1217 ± 0.0007 ^Ab^	0.1582 ± 0.0002 ^Ab^	0.1032 ± 0.0002 ^Bb^	0.1323 ± 0.0001 ^Bb^	0.0969 ± 0.0004 ^Cb^	0.1239 ± 0.0004 ^Cb^	0.0861 ± 0.0001 ^Db^	0.1150 ± 0.0001 ^Db^	0.0741 ± 0.0001 ^Eb^	0.1133 ± 0.0001 ^Db^			
	100	0.1257 ± 0.0002 ^Aa^	0.1638 ± 0.0001 ^Aa^	0.1228 ± 0.0002 ^Ba^	0.1534 ± 0.0001 ^Ba^	0.1082 ± 0.0002 ^Ca^	0.1532 ± 0.0007 ^Ca^	0.0969 ± 0.0001 ^Da^	0.1377 ± 0.0001 ^Da^	0.0888 ± 0.0001 ^Ea^	0.1194 ± 0.0005 ^Ea^			
Gly	0	0.0308 ± 0.0001 ^Ae^	0.0357 ± 0.0001 ^Ad^	0.0272 ± 0.0002 ^Be^	0.0333 ± 0.0001 ^Bd^	0.0267 ± 0.0002 ^BCe^	0.0295 ± 0.0004 ^Ce^	0.0241 ± 0.0004 ^Ce^	0.0287 ± 0.0001 ^De^	0.0210 ± 0.0001 ^Ce^	0.0225 ± 0.0001 ^Ee^	<0.01	**	**
	25	0.0334 ± 0.0001 ^Ad^	0.0354 ± 0.0001 ^Ad^	0.0333 ± 0.0003 ^Bd^	0.0339 ± 0.0001 ^Ad^	0.0310 ± 0.0001 ^Cd^	0.0334 ± 0.0001 ^Bd^	0.0278 ± 0.0001 ^Cd^	0.0306 ± 0.0001 ^Cd^	0.0263 ± 0.0004 ^Cd^	0.0285 ± 0.0001 ^Cd^			
	50	0.0372 ± 0.0001 ^Ac^	0.0390 ± 0.0001 ^Ac^	0.0366 ± 0.0001 ^Bc^	0.0379 ± 0.0001 ^Bc^	0.0350 ± 0.0001 ^Cc^	0.0368 ± 0.0001 ^Cc^	0.0321 ± 0.0007 ^CDc^	0.0363 ± 0.0001 ^Dc^	0.0297 ± 0.0002 ^Dc^	0.0291 ± 0.0001 ^Ec^			
	75	0.0469 ± 0.0001 ^Ab^	0.0510 ± 0.0001 ^Ab^	0.0447 ± 0.0001 ^ABb^	0.0413 ± 0.0001 ^Bb^	0.0422 ± 0.0001 ^Bb^	0.0384 ± 0.0007 ^Bb^	0.0395 ± 0.0003 ^BCb^	0.0375 ± 0.0001 ^Bb^	0.0356 ± 0.0005 ^Cb^	0.0360 ± 0.0002 ^Cb^			
	100	0.0614 ± 0.0001 ^Aa^	0.0564 ± 0.0001 ^Aa^	0.0534 ± 0.0002 ^Ba^	0.0527 ± 0.0006 ^Ba^	0.0498 ± 0.0002 ^Ca^	0.0490 ± 0.0003 ^BCa^	0.0457 ± 0.0007 ^Da^	0.0489 ± 0.0002 ^Ca^	0.0437 ± 0.0005 ^Da^	0.0422 ± 0.0001 ^Da^			
Ala	0	0.0411 ± 0.0001 ^Ad^	0.0463 ± 0.0001 ^Ae^	0.0377 ± 0.0003 ^Bc^	0.0450 ± 0.0001 ^Ae^	0.0371 ± 0.0001 ^Bc^	0.0413 ± 0.0001 ^Be^	0.0332 ± 0.0003 ^Cd^	0.0395 ± 0.0002 ^Ce^	0.0300 ± 0.0001 ^De^	0.0311 ± 0.0001 ^De^	<0.01	**	**
	25	0.0413 ± 0.0001 ^Ad^	0.0477 ± 0.0001 ^Ad^	0.0378 ± 0.0003 ^Bc^	0.0469 ± 0.0001 ^Ad^	0.0372 ± 0.0001 ^Bc^	0.0473 ± 0.0001 ^Ad^	0.0362 ± 0.0002 ^Cc^	0.0431 ± 0.0001 ^Bd^	0.0342 ± 0.0002 ^Dd^	0.0378 ± 0.0001 ^Cd^			
	50	0.0426 ± 0.0001 ^Ac^	0.0517 ± 0.0001 ^Ac^	0.0386 ± 0.0002 ^Bc^	0.0510 ± 0.0001 ^Ac^	0.0376 ± 0.0001 ^BCc^	0.0489 ± 0.0001 ^Bc^	0.0371 ± 0.0007 ^CDc^	0.0469 ± 0.0005 ^Cc^	0.0361 ± 0.0005 ^Dc^	0.0392 ± 0.0001 ^Dc^			
	75	0.0478 ± 0.0001 ^Ab^	0.0636 ± 0.0001 ^Ab^	0.0451 ± 0.002 ^Bb^	0.0566 ± 0.0003 ^Bb^	0.0429 ± 0.0006 ^Cb^	0.0506 ± 0.0001 ^Cb^	0.0404 ± 0.0005 ^Db^	0.0484 ± 0.0001 ^Db^	0.0385 ± 0.0008 ^Eb^	0.0475 ± 0.0001 ^Db^			
	100	0.0637 ± 0.0001 ^Aa^	0.0689 ± 0.0001 ^Aa^	0.0571 ± 0.0003 ^Ba^	0.0680 ± 0.0001 ^Aa^	0.0458 ± 0.0001 ^Ca^	0.0645 ± 0.0001 ^Aa^	0.0447 ± 0.0002 ^Ca^	0.0638 ± 0.0001 ^Aa^	0.0418 ± 0.0002 ^Da^	0.0521 ± 0.0004 ^Ba^			
Cys	0	0.0012 ± 0.0001 ^Bd^	0.0028 ± 0.0001 ^Ac^	0.0013 ± 0.0001 ^Bd^	0.0027 ± 0.0001 ^Ab^	0.0012 ± 0.0001 ^Bd^	0.0025 ± 0.0001 ^Ac^	0.0011 ± 0.0001 ^Be^	0.0023 ± 0.0001 ^Ab^	0.0052 ± 0.0061 ^Aa^	0.0008 ± 0.0001 ^Bc^	<0.01	**	**
	25	0.0017 ± 0.0001 ^Bc^	0.0032 ± 0.0001 ^Ab^	0.0016 ± 0.0001 ^Bc^	0.0027 ± 0.0001 ^Ab^	0.0015 ± 0.0001 ^Bc^	0.0026 ± 0.0001 ^Abc^	0.0013 ± 0.0001 ^Bd^	0.0024 ± 0.0001 ^Bb^	0.0064 ± 0.0075 ^Aa^	0.0011 ± 0.0001 ^Cc^			
	50	0.0019 ± 0.0001 ^Bc^	0.0033 ± 0.0001 ^Ab^	0.0018 ± 0.0001 ^Bc^	0.0029 ± 0.0001 ^ABb^	0.0018 ± 0.0001 ^Bc^	0.0027 ± 0.0001 ^ABabc^	0.0016 ± 0.0001 ^Bc^	0.0024 ± 0.0001 ^ABb^	0.0081 ± 0.0095 ^Aa^	0.0020 ± 0.0001 ^Bb^			
	75	0.0023 ± 0.0001 ^Bb^	0.0033 ± 0.0001 ^Ab^	0.0022 ± 0.0001 ^Bb^	0.0032 ± 0.0001 ^ABa^	0.0020 ± 0.0001 ^Bb^	0.0028 ± 0.0001 ^ABCab^	0.0018 ± 0.0001 ^Bb^	0.0024 ± 0.0001 ^BCb^	0.0089 ± 0.0104 ^Aa^	0.0022 ± 0.0001 ^Cb^			
	100	0.0039 ± 0.0001 ^Ba^	0.0036 ± 0.0001 ^Aa^	0.0034 ± 0.0001 ^BCa^	0.0032 ± 0.0001 ^ABa^	0.0030 ± 0.0001 ^BCa^	0.0029 ± 0.0001 ^ABa^	0.0026 ± 0.0001 ^Ca^	0.0027 ± 0.0001 ^ABa^	0.0129 ± 0.0150 ^Aa^	0.0025 ± 0.0001 ^Ba^			
Tyr	0	0.0252 ± 0.0001 ^ABe^	0.0780 ± 0.0001 ^Ae^	0.0248 ± 0.0001 ^Be^	0.0698 ± 0.0007 ^Be^	0.0264 ± 0.0001 ^Ad^	0.0674 ± 0.0004 ^Cd^	0.0221 ± 0.0001 ^Cd^	0.0629 ± 0.0002 ^De^	0.0192 ± 0.0003 ^De^	0.0600 ± 0.0002 ^Ee^	<0.01	**	**
	25	0.0286 ± 0.0002 ^Ad^	0.0856 ± 0.0001 ^Ad^	0.0277 ± 0.0001 ^Ad^	0.0768 ± 0.0028 ^Bd^	0.0267 ± 0.0001 ^Ac^	0.0847 ± 0.0014 ^Ac^	0.0232 ± 0.0002 ^Bc^	0.0687 ± 0.0002 ^Cd^	0.0216 ± 0.0005 ^Bd^	0.0635 ± 0.0002 ^Dd^			
	50	0.0300 ± 0.0001 ^Ac^	0.0921 ± 0.0001 ^Ac^	0.0291 ± 0.0001 ^Ac^	0.0882 ± 0.0015 ^ABc^	0.0274 ± 0.0001 ^Bb^	0.0859 ± 0.0004 ^ABc^	0.0267 ± 0.0001 ^Bb^	0.0749 ± 0.0006 ^ABc^	0.0236 ± 0.0003 ^Cc^	0.0712 ± 0.0015 ^Bc^			
	75	0.0326 ± 0.0001 ^Ab^	0.0993 ± 0.0001 ^Ab^	0.0324 ± 0.0001 ^Ab^	0.0928 ± 0.0018 ^Bb^	0.0277 ± 0.0002 ^Bb^	0.0907 ± 0.0001 ^Cb^	0.0269 ± 0.0001 ^Bb^	0.0877 ± 0.0002 ^Db^	0.0251 ± 0.0002 ^Cb^	0.0785 ± 0.0001 ^Eb^			
	100	0.0391 ± 0.0001 ^Aa^	0.1113 ± 0.0001 ^Aa^	0.0389 ± 0.0001 ^Aa^	0.1035 ± 0.0001 ^Ba^	0.0365 ± 0.0001 ^Ba^	0.0991 ± 0.0001 ^Ca^	0.0343 ± 0.0002 ^Ca^	0.0939 ± 0.001 ^Da^	0.0281 ± 0.0003 ^Da^	0.0871 ± 0.0008 ^Ea^			
Phe	0	0.0047 ± 0.0001 ^Ac^	0.0329 ± 0.0001 ^Ae^	0.0047 ± 0.0001 ^Ac^	0.0308 ± 0.0002 ^Be^	0.0045 ± 0.0001 ^Ac^	0.0297 ± 0.0002 ^Be^	0.0041 ± 0.0003 ^ABc^	0.0278 ± 0.0002 ^Cd^	0.0035 ± 0.0001 ^Bc^	0.0230 ± 0.0001 ^De^	<0.01	**	**
	25	0.0049 ± 0.0001 ^Ac^	0.0349 ± 0.0001 ^Ad^	0.0047 ± 0.0001 ^Ac^	0.0342 ± 0.0001 ^Ad^	0.0045 ± 0.0001 ^Ac^	0.0341 ± 0.0001 ^Ad^	0.0044 ± 0.0003 ^Abc^	0.0279 ± 0.0001 ^Bd^	0.0037 ± 0.0001 ^Bc^	0.0247 ± 0.0005 ^Cd^			
	50	0.0054 ± 0.0001 ^Ab^	0.0380 ± 0.0001 ^Ac^	0.0050 ± 0.0001 ^Bb^	0.0370 ± 0.0001 ^Ac^	0.0047 ± 0.0001 ^BCb^	0.0354 ± 0.0001 ^Bc^	0.0045 ± 0.0001 ^CDbc^	0.0317 ± 0.0001 ^Cc^	0.0042 ± 0.0001 ^Db^	0.0289 ± 0.0001 ^Dc^			
	75	0.0057 ± 0.0001 ^Aa^	0.0433 ± 0.0002 ^Ab^	0.0051 ± 0.0001 ^Bab^	0.0400 ± 0.0002 ^Bb^	0.0049 ± 0.0001 ^Bb^	0.0366 ± 0.0001 ^Cb^	0.0047 ± 0.0001 ^BCab^	0.0337 ± 0.0001 ^Db^	0.0043 ± 0.0002 ^Cb^	0.0321 ± 0.0004 ^Eb^			
	100	0.0058 ± 0.0001 ^Aa^	0.0480 ± 0.0001 ^Aa^	0.0052 ± 0.0001 ^Ba^	0.0448 ± 0.0001 ^Ba^	0.0052 ± 0.0001 ^Ba^	0.0412 ± 0.0002 ^Ca^	0.0050 ± 0.0001 ^Ba^	0.0350 ± 0.0001 ^Da^	0.0049 ± 0.0002 ^Ba^	0.0335 ± 0.0001 ^Ea^			
Lys	0	0.0006 ± 0.0001 ^Ac^	0.0718 ± 0.0001 ^Ae^	0.0005 ± 0.0002 ^Ad^	0.0715 ± 0.0004 ^Ae^	0.0006 ± 0.0001 ^Ac^	0.0652 ± 0.0001 ^Be^	0.0004 ± 0.0001 ^Ad^	0.0638 ± 0.0001 ^Ce^	0.0004 ± 0.0001 ^Ac^	0.0505 ± 0.0001 ^De^	<0.01	**	**
	25	0.0007 ± 0.0001 ^Ac^	0.0773 ± 0.0001 ^Ad^	0.0007 ± 0.0001 ^Acd^	0.0750 ± 0.0002 ^Bd^	0.0006 ± 0.0001 ^Ac^	0.0730 ± 0.001 ^Cd^	0.0005 ± 0.0001 ^Ad^	0.0702 ± 0.0002 ^Dd^	0.0005 ± 0.0001 ^Abc^	0.0601 ± 0.0002 ^Ed^			
	50	0.0013 ± 0.0001 ^Ab^	0.0812 ± 0.0001 ^Ac^	0.0010 ± 0.0001 ^ABbc^	0.0806 ± 0.0001 ^Ac^	0.0009 ± 0.0001 ^BCb^	0.0768 ± 0.0001 ^Bc^	0.0006 ± 0.0001 ^Cc^	0.0728 ± 0.0001 ^Cc^	0.0006 ± 0.0001 ^Cbc^	0.0637 ± 0.0001 ^Dc^			
	75	0.0013 ± 0.0001 ^Ab^	0.0958 ± 0.0001 ^Ab^	0.0012 ± 0.0002 ^Ab^	0.0898 ± 0.0003 ^Bb^	0.0010 ± 0.0001 ^ABb^	0.0801 ± 0.0001 ^Cb^	0.0008 ± 0.0001 ^Bb^	0.0747 ± 0.0002 ^Db^	0.0007 ± 0.0001 ^Bb^	0.0738 ± 0.0001 ^Db^			
	100	0.0019 ± 0.0001 ^Aa^	0.1056 ± 0.0001 ^Aa^	0.0019 ± 0.0002 ^Aa^	0.0983 ± 0.0002 ^ABa^	0.0015 ± 0.0001 ^Ba^	0.0985 ± 0.0002 ^Ba^	0.0012 ± 0.0001 ^BCa^	0.0960 ± 0.0002 ^Ba^	0.0010 ± 0.0001 ^Ca^	0.0787 ± 0.0001 ^Ca^			
His	0	0.0259 ± 0.0002 ^Ae^	0.0683 ± 0.0001 ^Ae^	0.0254 ± 0.0001 ^Ae^	0.0651 ± 0.0001 ^Be^	0.0230 ± 0.0001 ^Bd^	0.0612 ± 0.0003 ^Ce^	0.0218 ± 0.0001 ^Be^	0.0551 ± 0.0002 ^De^	0.0180 ± 0.0007 ^Cd^	0.0503 ± 0.0002 ^Ee^	<0.01	**	**
	25	0.0272 ± 0.0001 ^Ad^	0.0718 ± 0.0001 ^Ad^	0.0266 ± 0.0001 ^Ad^	0.0676 ± 0.0001 ^Bd^	0.0261 ± 0.0002 ^ABc^	0.0639 ± 0.0001 ^Cd^	0.0249 ± 0.0001 ^Bd^	0.0587 ± 0.0002 ^Dd^	0.0216 ± 0.0001 ^Cc^	0.0540 ± 0.0004 ^Ed^			
	50	0.0277 ± 0.0001 ^Ac^	0.0789 ± 0.0001 ^Ac^	0.0271 ± 0.0001 ^ABc^	0.0726 ± 0.0007 ^Bc^	0.0264 ± 0.0001 ^Bc^	0.0681 ± 0.0002 ^Cc^	0.0253 ± 0.0001 ^Cc^	0.0646 ± 0.0004 ^Dc^	0.0222 ± 0.0001 ^Dc^	0.0601 ± 0.0004 ^Ec^			
	75	0.0348 ± 0.0002 ^Ab^	0.0881 ± 0.0002 ^Ab^	0.0317 ± 0.0001 ^Bb^	0.0792 ± 0.0001 ^Bb^	0.0270 ± 0.0001 ^Cb^	0.0698 ± 0.0001 ^Cb^	0.0268 ± 0.0001 ^Cb^	0.0683 ± 0.0002 ^Db^	0.0250 ± 0.0006 ^Db^	0.0644 ± 0.0003 ^Eb^			
	100	0.0369 ± 0.0001 ^Aa^	0.0970 ± 0.0002 ^Aa^	0.0331 ± 0.0001 ^Ba^	0.0873 ± 0.0001 ^Ba^	0.0320 ± 0.0002 ^BCa^	0.0788 ± 0.0001 ^Ca^	0.0318 ± 0.0001 ^Ca^	0.0757 ± 0.0016 ^Da^	0.0281 ± 0.0001 ^Da^	0.0724 ± 0.001 ^Ea^			
Pro	0	0.0245 ± 0.0001 ^Ae^	0.0280 ± 0.0001 ^Ad^	0.0214 ± 0.0001 ^Be^	0.0256 ± 0.0001 ^Be^	0.0188 ± 0.0001 ^Cd^	0.0231 ± 0.0001 ^Cd^	0.0181 ± 0.0002 ^Ce^	0.0230 ± 0.0001 ^Ce^	0.0153 ± 0.0001 ^De^	0.0184 ± 0.0001 ^De^	<0.01	**	**
	25	0.0262 ± 0.0001 ^Ad^	0.0287 ± 0.0001 ^Ad^	0.0223 ± 0.0001 ^Bd^	0.0268 ± 0.0001 ^Bd^	0.0220 ± 0.0002 ^Bc^	0.0268 ± 0.0001 ^Bc^	0.0192 ± 0.0001 ^Cd^	0.0253 ± 0.0001 ^Cd^	0.0170 ± 0.0001 ^Dd^	0.0220 ± 0.0002 ^Dd^			
	50	0.0270 ± 0.0001 ^Ac^	0.0323 ± 0.0005 ^Ac^	0.0269 ± 0.0008 ^Ac^	0.0307 ± 0.0001 ^Bc^	0.0225 ± 0.0001 ^Bc^	0.0303 ± 0.0001 ^Bb^	0.0211 ± 0.0001 ^BCc^	0.0281 ± 0.0001 ^Cc^	0.0195 ± 0.0001 ^Cc^	0.0235 ± 0.0002 ^Dc^			
	75	0.0348 ± 0.0001 ^Ab^	0.0369 ± 0.0002 ^Ab^	0.0320 ± 0.0001 ^Bb^	0.0321 ± 0.0001 ^Bb^	0.0274 ± 0.0001 ^Cb^	0.0301 ± 0.0001 ^Cb^	0.0242 ± 0.0002 ^Db^	0.0294 ± 0.0001 ^Cb^	0.0225 ± 0.0001 ^Eb^	0.0293 ± 0.0002 ^Cb^			
	100	0.0447 ± 0.0001 ^Aa^	0.0416 ± 0.0001 ^Aa^	0.0403 ± 0.0001 ^Ba^	0.0408 ± 0.0007 ^Aa^	0.0369 ± 0.001 ^Ca^	0.0379 ± 0.0001 ^Ba^	0.0292 ± 0.0002 ^Da^	0.0378 ± 0.0001 ^Ba^	0.0274 ± 0.0005 ^Ea^	0.0323 ± 0.0001 ^Ca^			

Note: Data are expressed as mean ± standard deviation. Upper case letters indicate significant differences in storage times. Lower case letters indicate significant differences in fat substitution rates. S.E: Standard error. n.s: Not significant. R: substitution rate. T: Cooling time. I. Interaction. ** *p* < 0.01.

## Data Availability

The data generated from the study is clearly presented and discussed in the manuscript.

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
