# Peer review of "Preparation of Bovine Hides Gelatin by Ultra-High Pressure Technique and the Effect of Its Replacement Fat on the Quality and In Vitro Digestion of Beef Patties"

_foods, 2023, doi:10.3390/foods12163092_

Round 1

Reviewer 1 Report

Comments and Suggestions for Authors

Line 2: ultra-high ? or ultrahigh (standardize trough the manuscript)

Line 3,18: use italic text format for in vitro (modify through the document)

Line 15: 0-4 or 0–4 like in line 134

Line 21,23: use the meaning of the abbreviations

Line 39: insert space…. [4]. Therefore

Line 57: insert space…. [14]. To

Line 62,75,81,etc: avoid using the names of the authors, it is only recommended to number the reference in parentheses with the number that corresponds to it (correct through the document)

Line 73: Ultra-high pressure (UPH) technology…. Note: from here on, use only the abbreviation in the text

Line 102: did you mean? ….company

Line 103: What were the conditions for degreasing?

Line 112: insert a dot…. 2.2.

Line 114: 5 or five like in line 115 for three

Line 117: Could include information on the vacuum packer used (model, brand, country)

Line 119: Could include information on the UHP used (model, brand, country)

Line 121: insert space…. device.Both

Line 124: insert a dot…. 2.3.

Line 125: include the pH values of the sample used

Line 127,131: include trademark of the equipment

Line 133: include information of the equipment, fryer (model, brand, country)

Line 136: g or g/100g… delete (g) from Item

Line 137: insert a dot…. 2.4.

Line 156: insert space…. 2. There

Line 164: ….. The

Line 171: include information of the equipment, texture analyzer (model, brand, country)

Line 162: insert a dot…. 2.5.

Line 168: insert a dot…. 2.6.

Line 175: insert a dot…. 2.7.

Line 177: … according to 2.3?? 2.3 section

Line 184: insert a dot…. 2.8.

Line 189: include information of the equipment, mixer (model, brand, country)

Line 190: include information of the equipment, water bath (model, brand, country)

Line 192: include information of the equipment, centrifuge (model, brand, country)

Line 193: r or rpm?

Line 199: insert a dot…. 2.9.

Line 207: include information of the equipment, oven (model, brand, country)

Line 207: modify…. 22–24 h

Line 208: ….after the HCl was fully…

Line 211: insert a dot…. 2.10.

Line 217: mg) and

Line 218: to the

Line 219: 7.5 ± 0.1

Line 225: include information of the equipment, freeze dryer (model, brand, country)

Line 237: insert a dot…. 3.1.

Line 239: insert spaces…. Hardness (N)…. pressure (MPa)….. springiness (mm)….

Line 240: Figure 1.

Line 266: insert spaces…. Hardness (N)…. Time (min)… loss (%)…. Elasticity (mm)…. springiness (mm)….

Line 267: Figure 2.

Line 276: 20 min. Cooking…

Line 288: Figure 3.

Line 288: correct…. NaCl

Line 288: insert spaces…. use spaces between the text and the unit where required in the image

Line 289: correct…. NaCl

Line 295: 1–1.5%

Line 299: 1–1.7%

Line 311: the figure must appear after being mentioned in the text

Line 311: Figure 4.

Line 311: insert spaces…. use spaces between the text and the unit where required in the image

Line 335: the tables must appear after being mentioned in the text

Line 347: insert superscript text format for R2

Line 358: p < 0.05

Line 374: (Figure 5)

Line 388: Figure 6. Validation

Line 398: …. in Figure 3

Line 401: insert a dot…. 3.2.

Line 406,407: some terms were previously abbreviated, just include the abbreviation

Line 411: …. changes in PUFA due….

Line 419: SFA [43].

Line 421: comparing

Line 422: Could you abbreviate the term simulated gastrointestinal digestion (sGD) throughout the document?

Line 432: (Figure 7A)

Line 435: (Figure 7B)

Line 436: 2 SFA?

Line 437: 4 MUFA?

Line 437: 2 PUFA?

Line 444: human health. ?

Line 447: p < 0.05

Line 447: p < 0.01

Line 449: Figure 7. (A)

Line 452: insert a dot…. 3.3.

Line 459: compared?

Line 475: (Figure 8B)

Line 486: Figure 8. (A)

Line 493: insert a dot…. 3.4.

Line 493: analysis

Line 494: Figure 9. (A)

Line 505: Figure 10. (A)

Note: insert a dot at the end of each figure and table title

Line 517: insert a dot…. 3.5.

Line 518: Figure 11. (A)

Line 524: Figure 11

Line 529: PC1. PC2

Line 539: Figure 12. (A)

Line 558: insert a dot…. 3.6.

Line 559: Figure 13.

Line 558: insert spaces…. use spaces between the text and the unit where required in the image

Line 580: gelatin. The

Line 604,609,612,614, correct trough the references: use the abbreviated format for the journal name of the reference

Line 606,607,609,etc: use the correct text format for the title of the reference

Line 608,617, correct trough this section: when using the abbreviation of the journal name of the reference, a dot must be used

Line 608: S1–36

Line 610: C2183–C2188

Line 612: 113–125

Line 614: S543–S555

Line 617: 368–377 (use – instead of - , correct trough this section

Line 619,629,etc: LWT

Was the effect on the pH and oxidation of the elaborated hamburgers measured?

Reviewer 2 Report

Comments and Suggestions for Authors

Dear Authors

In my opinion, this manuscript needs to be revised as follows:

1- Accurate and correct keyword selection is one of the most important ways to improve and expand the searchability of the article after publication. Often the words used in the title and keywords are the same. Authors should use other keywords as manuscript keywords.

2- In line 13 (abstract), you mentioned: "..with beef skin collagen...", while the title is: "Preparation of bovine hide gelatin by....", gelatin is not collagen, gelatin is obtained from collagen by chemical, enzymatic and thermal treatments, please check that.

3- Why did you choose 28 days for evaluation?

4- Please enrich the introduction section by using these articles as new references:

- https://doi.org/10.3390/foods12030670

-https://doi.org/10.1016/j.foodhyd.2022.107503 

5- In your opinion, which feature is the novelty of this work, please mention it in the last paragraph of the introduction.

6- Results, discussions, Figures, and Tables are proper and qualified.

Bests,

Comments on the Quality of English Language

Please check the manuscript by a native English speaker.

Author Response

Thank you very much! Please refer to the attachment.

Reviewer 3 Report

Comments and Suggestions for Authors

This paper deals with addition of bovine hides gelatin as a replacement for fat on the quality and in vitro digestion of beef patties. Although the topic is interesting and has some scientific merit, the presentation of  the results is very extensive and had to follow. I suggest rewriting the whole paper, shortening it by the half. In this form the paper cannot be accepted for publication.

Round 2

Reviewer 2 Report

Comments and Suggestions for Authors

Accept!

Author Response

Dear reviewer:

We would like to express our sincere gratitude to you for taking out your valuable time to revise our paper and for your constructive comments to make our article of better quality! Thank you again for your support in revising my revised content. Once again, we appreciate your support.

Thank you, with most sincere regards.

Yours sincerely.

Mengying Liu 

Corresponding author: Li Zhang

Reviewer 3 Report

Comments and Suggestions for Authors

I still thing this manuscript is too long. I leave the final decision the Editor.

Author Response

(The authors gave the same response as above.)
